# Geometric Spatiotemporal Transformer to Simulate Long-Term Physical Dynamics

## Abstract

Physical dynamics simulation plays a crucial role in various real-world applications. In this paper, we explore the potential of leveraging Transformers by framing the task as autoregressive next-graph prediction based on spatiotemporal graph inputs. To achieve this, we propose Geometric Spatiotemporal Transformers (GSTs), which adopt the expressive encoder-decoder architecture of traditional Transformers. At the core of GSTs are equivariant spatiotemporal blocks that alternate between spatial and temporal modules while preserving E(3) symmetries. Additionally, we introduce the Temporal Difference Graph (TDG), derived from the difference between the last two frames of historical input, to capture global dynamic patterns and mitigate cumulative errors in long-term prediction tasks. Unlike existing Graph Neural Network (GNN) methods, GSTs can process full input sequences of arbitrary lengths to effectively capture long-term context, and address cumulative errors over long-term rollouts thanks to the TDG mechanism. Our method achieves state-of-the-art performance across multiple challenging physical systems at various scales (molecular-, protein-, and macro-level), demonstrating the robust dynamics simulation capabilities.

## 1 Introduction

Accurately simulating the dynamics of physical systems forms the cornerstone of numerous applications. For example, in drug discovery, molecular dynamics simulations provide profound insights into the binding interactions between drug molecules and their target proteins (Salo-Ahen et al., 2020). Plenty of methods (Battaglia et al., 2016; Sanchez-Gonzalez et al., 2020) have emerged to simulate physical dynamics as graph translation via Graph Neural Networks (GNNs), given that many physical systems can be effectively represented as graphs. Further advancements have been made by leveraging geometric GNNs (Satorras et al., 2021b; Fuchs et al., 2020; Huang et al., 2022), which ensure the dynamics to be independent to any rotation, reflection, or translation transformations, thereby aligning seamlessly with E(3) symmetries inherent in physics. Building upon geometric GNNs, several studies (Xu et al., 2023; Wu et al., 2024) adopt a spatiotemporal approach, rather than the previous frame-to-frame setting, which leverages multiple frames to predict the next one, thereby capturing long-term historical information and recovering non-Markovian interactions.

In this work, we investigate the potential of Transformers (Vaswani, 2017) for simulating graph-based physical dynamics. While the Transformer architecture has become the de facto standard in Natural Language Processing (NLP) and various other domains, its application in physical dynamics simulations—particularly in graph-based contexts—remains underexplored. Given that both natural languages and physical trajectories are sequential data, it is promising to leverage the success of Transformers for physical dynamics representation and generation. However, notable differences exist between the two tasks. From a data structure perspective, we need to process spatiotemporal graphs rather than text sequences, and the objective shifts from next-token prediction to next-graph prediction. Importantly, the model must conform to certain physical rules, such as E(3) symmetries, to ensure generalizability across arbitrary coordinate systems. Additionally, addressing cumulative errors is crucial for long-term simulations, necessitating specific design considerations for the model.

To effectively bridge these gaps, we propose Geometric Spatiotemporal Transformers (GSTs). By inheriting the encoder-decoder architecture from the original Transformer, GSTs can accept spatiotemporal inputs of arbitrary temporal length and predict long-term future graphs in an autoregres-

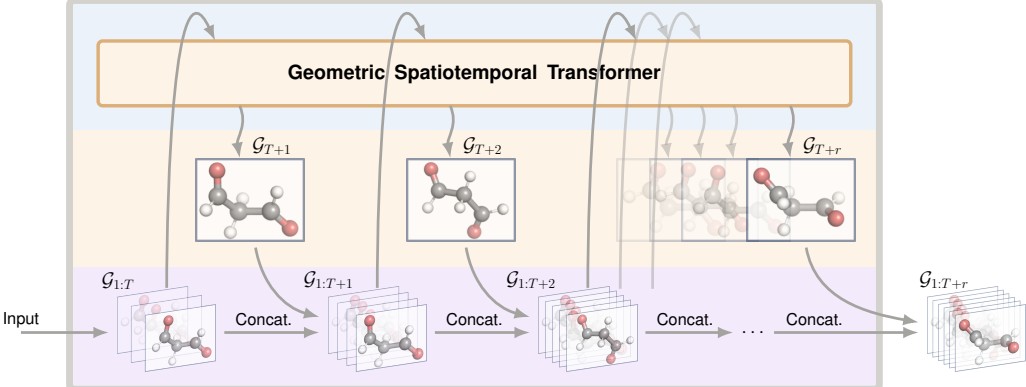

Figure 1: Illustration of how Geometric Spatiotemporal Transformers (GST) work on molecular dynamics. It processes full spatiotemporal inputs and predict future frames autoregressively.

sive manner, as illustrated in Fig. 1. The core of GST consists of E(3)-equivariant spatiotemporal blocks, which alternate between spatial and temporal modules while preserving E(3) symmetries. More importantly, to circumvent the issue of cumulative errors, we predict the difference from the last frame rather than predicting the next frame directly, as estimating the difference between the current frame and the next may be easier than making a direct prediction of the next frame. To achieve this, we introduce a Temporal Difference Graph (TDG). Initialized as the difference between the last two frames in the input layer, the TDG interacts with all other frames to gather global dynamical patterns in the following layers and and is used alongside the last frame for next-frame prediction. The encoder takes as input the initial trajectory and all predicted frames, while the decoder's input is further augmented with the TDG.

In contrast to existing methods (Satorras et al., 2021b; Wu et al., 2024), our GST exhibits several crucial benefits. On one hand, as a derivative of the Transformer architecture, GST inherits its strong expressivity and allows for more flexible settings. GST can process full input sequences of variable lengths to capture long-term context or adapt to fixed-length inputs for efficiency. In contrast, previous methods are limited to fixed-length input settings (Wu et al., 2024) or even one-frame inputs (Satorras et al., 2021b). On the other hand, while current works primarily focus on predicting a single frame (Wu et al., 2024), we are concerned with estimating long-term trajectories to facilitate practical applications, which also confronts the cumulative error issue. To address this problem, we leverage the TDG to focus more on the temporal difference prediction and perform model training under a long-term autoregressive loss.

In summary, the contributions of this paper can be summarized as follows:

- We propose GST, a novel Transformer to simulate long-term physical dynamics autoregressively. GST inherits the strong expressivity and flexible designs from original Transformers, while promisingly respecting the spatiotemporal geometries and E(3) symmetries.
- We define TDG to reduce the impact of cumulative errors in long-term rollouts. Initialized as the difference between the last two frames, the TDG interacts with all other frames to capture global dynamics and is used alongside the last frame for next-frame prediction.
- We conduct extensive experiments on real-world datasets across three different scales: molecules, proteins, and human motions. Results demonstrate that our model achieves superior performance across various challenging settings and exhibits a larger gap in longer-horizon rollout tasks compared to existing State-Of-The-Art (SOTA) models.

## 2 RELATED WORK

**Physical Dynamics Simulation.** The simulation of physical dynamics has garnered significant attention due to its widespread applications in real-world scenarios. Early work (Wu et al., 2015;

2017) combines physics engines with graphics engines, supplemented by deep learning techniques, to achieve preliminary understanding and modeling of physical dynamics. Later the Interaction Network (IN) (Battaglia et al., 2016) is proposed to employ GNN-like message passing to model object interactions in physical systems. Inspired by IN, researchers increasingly recognized GNNs as powerful tools for modeling physical dynamics, owing to their inherent ability to capture complex relational structures. This realization spawned numerous extensions and improvements (Li et al., 2020; Finzi et al., 2020). Some approaches also integrated ordinary differential equations with GNNs to model complex dynamics (Sanchez-Gonzalez et al., 2019). However, these methods often overlooked the crucial role of physical symmetries. Consequently, a series of equivariant GNN models encoding geometric information rapidly emerge (Thomas et al., 2018; Fuchs et al., 2020; Huang et al., 2022). Despite these advancements, existing methods frequently suffer from constraints such as fixed-length input requirements (Xu et al., 2023; Wu et al., 2024) or one-frame input limitations (Satorras et al., 2021b). Moreover, many focus solely on single-frame predictions (Han et al., 2022). In contrast, this paper presents a novel model capable of flexibly accommodating variable-length or fixed-length inputs. Furthermore, we explore its performance on long-term trajectory prediction tasks, addressing a critical gap in the current literature.

**Deep Spatiotemporal Models.** Deep spatiotemporal models have gained prominence across diverse real-world applications (Cini et al., 2024; Marisca et al., 2024; Li et al., 2024). In traffic prediction, STGCN (Yu et al., 2017) and DCRNN (Li et al., 2017) leverage graph structures with recurrent or convolutional layers. Attention-based architectures like Gaan (Zhang et al., 2018) and AGL-STAN (Sun et al., 2022) capture temporal and spatial dependencies for traffic flow forecasting and crime prediction. Transformer-based models such as MMST-ViT Lin et al. (2023), Multi-SPANS(Zou et al., 2024), and MOIRAI (Woo et al., 2024) have been applied to various Spatiotemporal tasks including climate change-aware crop yield prediction, traffic forecasting, and general time-series analysis. While effective in their domains, these models typically lack consideration for 3D spatial symmetries inherent in physical systems, limiting their applicability to physical dynamics simulation. Addressing this, we adapt the standard Transformer architecture to 3D physical dynamics simulation tasks. By introducing equivariant designs, we ensure conformity with E(3) symmetries, bridging the gap between general spatiotemporal modeling and physical dynamics simulation.

## 3 OUR METHOD

In this section, we first introduce the necessary preliminaries related to physical dynamics simulation. Then, we describe the framework of our model, which adopts an equivariant encoder-decoder architecture, along with a temporal difference graph to reduce the cumulative errors in long-term predictions. Fig. 2 illustrates the overall framework of our model.

### 3.1 NOTATIONS AND DEFINITIONS

The trajectory of a physical system (*e.g.* a molecule) over a temporal length $T$ and with a time lag of $\Delta t = 1$[1], can be modeled as a spatiotemporal graph $\mathcal{G}_{1:T} := \{\mathcal{G}_t = (\mathcal{V}_t, \mathcal{E})\}_{t=1}^T$, where different frame $\mathcal{G}_t$ shares the same node identities (*e.g.* atoms) and edge connections (*e.g.* bonds), and the $i$-th node $v_{t,i}$ at time $t$ is associated with an invariant feature $\boldsymbol{h}_{t,i} \in \mathbb{R}^c$ (*e.g.* atom types) and an equivariant 3D coordinate vector $\vec{\boldsymbol{x}}_{t,i} \in \mathbb{R}^3$. Particularly for $\boldsymbol{h}_t$, we further add temporal position embedding with the sine function. We do not leverage edge features in this work. We hereafter denote by the matrices $\boldsymbol{H}_t$ and $\vec{\boldsymbol{X}}_t$ the collection of all node features and coordinates in $\mathcal{G}_t$.

**Task Formulation.** As illustrated in Fig. 1, given the observed trajectory $\mathcal{G}_{1:T}$, our goal is to learn a function $\phi$ that autoregressively predicts the long-term future frames $G_{T+1:T+R}$ over a duration of $R \gg 1$. This process can be formally expressed as follows:

$$\mathcal{G}_{T+1} = \phi(\mathcal{G}_{1:T}), \ldots, \mathcal{G}_{T+r} = \phi(\mathcal{G}_{1:T+r-1}), \ldots, \mathcal{G}_{T+R} = \phi(\mathcal{G}_{1:T+R-1}). \tag{1}$$

The above autoregressive prediction is also called the rollout process in the domain. In practice, we only require to predict the 3D coordinates $\vec{\boldsymbol{X}}_{T+r}$, while the corresponding invariant features $\boldsymbol{h}_{T+r}$ can be computed manually.

---

[1]Here $\Delta t$ is chosen as 1 for simplicity, while it can be selected remarkably larger than 1 for the acceleration of dynamics simulations in practice.

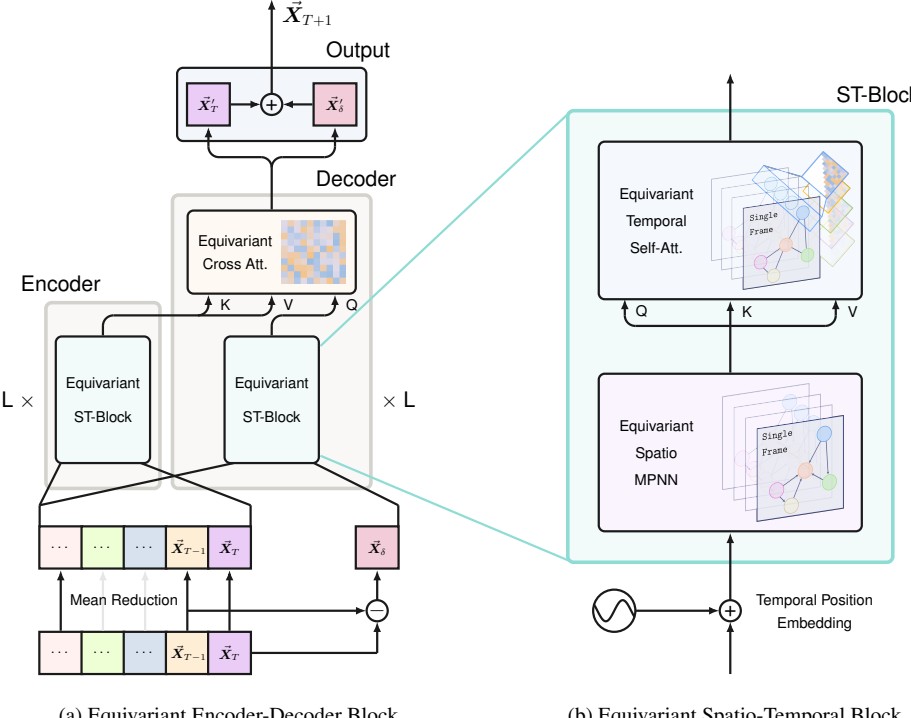

(a) Equivariant Encoder-Decoder Block      (b) Equivariant Spatio-Temporal Block

Figure 2: The overall architecture of GST. Initially, a "Mean Reduction" operation is applied to the historical trajectory $\vec{X}_1, ..., \vec{X}_T$. The entire historical trajectory is then input into an equivariant encoder. After initializing $\vec{X}_\delta$ with the last two frames, we concatenate the historical trajectory with $\vec{X}_\delta$ to form the input for the equivariant decoder. Temporal and spatial dependencies are captured using Equivariant ST-Blocks and Equivariant Cross-attention modules. The final updated $\vec{X}_T^{'}$ is summed with $\vec{X}_\delta^{'}$ to generate the coordinates for the subsequent frame, $\vec{X}_{T+1}$.

**Equivariance.** An important inductive bias to consider is that the function $\phi$ should be E(3)-equivariant, ensuring that the dynamics remain independent of the observation perspective. This means that if the input trajectory undergoes any arbitrary translation, reflection, or rotation transformation, the output of $\phi$ should undergo a corresponding transformation.

**Comparisons with Previous Settings.** One remarkable benefit of our function defined in Eq. (1) is its ability to accommodate inputs of variable temporal length, enhancing flexibility for various applications. It can take all the historical frames as input for modeling the complete context as defined by default; in case of efficiency considerations, it can also be modified to accept only fixed-length inputs, which degenerates to $\tilde{\mathcal{G}}_{T+r} = \phi(\mathcal{G}_{T+r-L:T+r-1})$ using only previous $L$ frames to predict the next frame. On the contrary, previous methods such as EGNN (Satorras et al., 2021a), GNS (Sanchez-Gonzalez et al., 2020), and ESTAG (Wu et al., 2024) can only admit the fixed-length input setting. Furthermore, our task emphasizes long-term dynamics prediction, whereas existing methods mostly focus on predicting only one future frame. The architecture of $\phi$ is specifically designed and will be elaborated on in the following subsections.

## 3.2 THE PROPOSED GST MODEL

Inspired by the original Transformer framework (Vaswani, 2017), the architecture of our proposed GST also consists of an encoder and a decoder, as displayed in Fig. 2. To predict $G_{T+r}$, the encoder takes as input the initial trajectory and all predicted frames, namely $\mathcal{G}_{1:T+r-1}$. The decoder's input is further augmented with an artificial graph $\mathcal{G}_\delta$, forming the input $(\mathcal{G}_{1:T+r-1}, \mathcal{G}_\delta)$. Both the encoder and the decoder employ a certain number of equivariant spatiotemporal blocks for representation learning. The output from the encoder is integrated into the decoder through an equivariant cross-attention layer. Further details are provided below.

(1) **Equivariant Encoder**

The encoder is comprised of $L$ equivariant spatiotemporal blocks. For each block, it alternates two modules: equivariant spatial message passing and equivariant temporal self-attention passing. We denote the $t$-th frame of the $l$-th layer as $\mathcal{G}_t^{e,l}(\boldsymbol{h}_t^{e,l}, \vec{\boldsymbol{X}}_t^{e,l})$.

The spatial module leverages EGNN (Satorras et al., 2021a) to characterize the spatial geometry of each frame individually, which is formally delineated as:

$$\boldsymbol{m}_{t,ij}^{e,l} = \varphi_m \left( \boldsymbol{h}_{t,i}^{e,l}, \boldsymbol{h}_{t,j}^{e,l}, \left\| \vec{\boldsymbol{x}}_{t,i}^{e,l} - \vec{\boldsymbol{x}}_{t,j}^{e,l} \right\| \right),$$

$$\boldsymbol{h}_{t,i}^{e,l+1} = \boldsymbol{h}_{t,i}^{e,l} + \varphi_h \left( \boldsymbol{h}_{t,i}^{e,l}, \sum_{j \in \mathcal{N}(i)} \boldsymbol{m}_{t,j}^{e,l} \right), \tag{2}$$

$$\vec{\boldsymbol{x}}_{t,i}^{e,l+1} = \vec{\boldsymbol{x}}_{t,i}^{e,l} + \frac{1}{|\mathcal{N}(i)|} \sum_{j \in \mathcal{N}(i)} \varphi_x \left( \boldsymbol{m}_{t,ij}^{e,l} \right) \cdot \left( \vec{\boldsymbol{x}}_{t,i}^{e,l} - \vec{\boldsymbol{x}}_{t,j}^{e,l} \right),$$

where $\varphi_m$, $\varphi_x$ and $\varphi_h$ are Multi-Layer Perceptrons (MLPs), $\mathcal{N}(i)$ represents all neighboring nodes of the $i$-th node. Particularly, $\boldsymbol{m}_{t,ij}^{e,l}$ is an E(3)-invariant message from node $j$ to $i$, which can be used to aggregate and update $\boldsymbol{h}_{t,i}^{e,l}$ features via $\varphi_h$; as for the update of $\vec{\boldsymbol{x}}_{t,i}^{e,l}$, $\varphi_x$ is used to compute a 1D scalar $\varphi_x(\boldsymbol{m}_{t,ij}^{e,l})$ which is then left-multiplied with $\vec{\boldsymbol{x}}_{t,i}^{e,l} - \vec{\boldsymbol{x}}_{t,j}^{e,l}$ to preserve directional information, and the residual connections are added to allow translation equivariance.

The temporal module employs an equivariant self-attention mechanism to model inter-frame dependencies and dynamical patterns for each node separately. One crucial benefit of using the attention strategy is that it can naturally processes inputs of different lengths, which perfectly fits our goal. Notably, in contrast to the conventional full-attention mechanism employed in traditional Transformer's encoder, we adopt a causal attention paradigm to preserve temporal consistency. The efficacy of this causal attention strategy will be empirically validated through ablation studies. The temporal message passing can be formally characterized as follows:

$$\left. \begin{aligned} \boldsymbol{q}_{t,i}^{e,l+1} &= \varphi_{qe}(\boldsymbol{h}_{t,i}^{e,l+1}) \\ \boldsymbol{k}_{s,i}^{e,l+1} &= \varphi_{ke}(\boldsymbol{h}_{s,i}^{e,l+1}) \\ \boldsymbol{v}_{s,i}^{e,l+1} &= \varphi_{ve}(\boldsymbol{h}_{s,i}^{e,l+1}) \end{aligned} \right\} \implies \alpha_{ts,i}^{e,l+1} = \texttt{Softmax}\left( \left\langle \boldsymbol{q}_{t,i}^{e,l+1}, \boldsymbol{k}_{s,i}^{e,l+1} \right\rangle \right), \tag{3}$$

where $\varphi_q$, $\varphi_k$, $\varphi_v$, $\varphi_{xt}$ and $\varphi_{ht}$ are all MLPs. The invariant features $\boldsymbol{q}_{t,i}$, $\boldsymbol{k}_{s,i}$ and $\boldsymbol{q}_{s,i}$ refer to the query, key and value features, respectively, and $\alpha_{ts,i}^{e,l+1}$ signifies the attention weight between the $t$-th and $s$-th frames. With the derived attentions, we update the features by:

$$\boldsymbol{h}_{t,i}^{e,l+2} = \boldsymbol{h}_{t,i}^{e,l+1} + \varphi_{ht} \left( \boldsymbol{h}_{t,i}^{e,l+1}, \sum_{s=1}^{t} \alpha_{ts,i}^{e,l+1} \boldsymbol{v}_{s,i}^{e,l+1} \right),$$

$$\vec{\boldsymbol{x}}_{t,i}^{e,l+2} = \vec{\boldsymbol{x}}_{t,i}^{e,l+1} + \sum_{s=1}^{t} \alpha_{ts,i}^{e,l+1} \cdot \varphi_{xt} \left( \boldsymbol{v}_{s,i}^{e,l+1} \right) \cdot \left( \vec{\boldsymbol{x}}_{t,i}^{e,l+1} - \vec{\boldsymbol{x}}_{s,i}^{e,l+1} \right). \tag{4}$$

Consequently, the encoder yields a refined and compressed representation of the sequential features, encapsulating both spatial and temporal dependencies in a more compact and informative format.

(2) **Equivariant Decoder**

Similar to the encoder, our decoder also applies $L$ equivariant spatiotemporal blocks. The main difference is that the decoder processes not only the spatiotemporal graph but also the TDG $\mathcal{G}_\delta$. Below we first define the concept of the TDG in prior to the introduction of the decoder architecture.

**Temporal Difference Graph** $\mathcal{G}_\delta$**.** Existing methods tend to amplify cumulative errors as the rollout distance increases. We circumvent this issue by basing each prediction solely on the combination of the last predicted frame and $\mathcal{G}_\delta$, significantly reducing error propagation in long-term predictions. This approach is motivated by the strong local correlation observed between adjacent frames in physical dynamics. Predicting the difference between the current frame and the next one may be

easier than directly predicting the next frame. To do so, we first initialize TDG as $\mathcal{G}_\delta = \mathcal{G}_{T+r-1} - \mathcal{G}_{T+r-2}$, namely, for each node $i$,

$$\boldsymbol{h}_{\delta,i} = \boldsymbol{h}_{T+r-1,i} - \boldsymbol{h}_{T+r-2,i}, \qquad\qquad \vec{\boldsymbol{x}}_{\delta,i} = \vec{\boldsymbol{x}}_{T+r-1,i} - \vec{\boldsymbol{x}}_{T+r-2,i}. \tag{5}$$

After the initialization, $\mathcal{G}_\delta$ along with $\mathcal{G}_{1:T+r-1}$ will be fed into the decoder. Within the decoder, $\mathcal{G}_\delta$ is considered as a global graph which interacts with each other graph to gather the global information to refine its node features layer by layer.

We now present the formulation of the spatiotemporal block. For conciseness, we denote the $t$-th frame (including the TDG frame) of the $l$-th layer as $\mathcal{G}_t^{d,l}(\boldsymbol{h}_t^{d,l}, \vec{\boldsymbol{X}}_t^{d,l})$. The spatial module follows the same mechanism as descried in Eq. (2) for all frames, including the TDG. In contrast, the temporal module performs causal attention (Eqs. (3) and (4)) among all frames, excluding the TDG. In particular, the update of the TDG in the temporal module is given by:

$$\boldsymbol{h}_{\delta,i}^{d,l+2} = \boldsymbol{h}_{\delta,i}^{d,l+1} + \varphi_{ht}\left(\boldsymbol{h}_{\delta,i}^{d,l+1}, \sum_{s=1}^{T+r-1} \alpha_{\delta s,i}^{d,l+1}\boldsymbol{v}_{s,i}^{d,l+1}\right) \cdot$$

$$\vec{\boldsymbol{x}}_{\delta,i}^{d,l+2} = \vec{\boldsymbol{x}}_{\delta,i}^{d,l+1} + \sum_{s=1}^{T+r-1} \alpha_{\delta s,i}^{d,l+1} \cdot \varphi_{xt}\left(\boldsymbol{v}_{s,i}^{d,l+1}\right) \cdot \left(\vec{\boldsymbol{x}}_{\delta,i}^{d,l+1} - (\vec{\boldsymbol{x}}_{s,i}^{d,l+1} - \bar{\boldsymbol{x}})\right), \tag{6}$$

where $\alpha_{\delta s,i}^{d,l+1}$ stands for the attention weight between $\mathcal{G}_\delta$ and the $s$-th frame. Importantly, since $\vec{\boldsymbol{x}}_{\delta,i}^{d,l+2}$ should be translation invariant, we have carried out mean reduction $\vec{\boldsymbol{x}}_{s,i}^{d,l+1} - \bar{\boldsymbol{x}}$ beforehand for the update of $\vec{\boldsymbol{x}}_{\delta,i}^{d,l+2}$, where $\bar{\boldsymbol{x}}$ is the mean of all nodes across all frames in $\mathcal{G}_{1:T}$.

Finally, we employ a cross-attention mechanism, utilizing $(\mathcal{G}_{1:T+r-1}^{d,l+2}, \mathcal{G}_\delta^{d,l+2})$ as the query to extract useful information from the encoder-compressed sequence $\mathcal{G}_{1:T+r-1}^{e,l+2}$. This formulation enables the model to dynamically focus on relevant historical information, thereby enhancing the fidelity and contextual relevance of spatiotemporal representations. This mechanism potentially leads to improved model performance and generalization capabilities in capturing complex spatiotemporal dynamics. The update procedure can be formally expressed as follows:

$$\left.\begin{array}{l} \boldsymbol{q}_{t,i}^{d,l+2} = \varphi_{qd}(\boldsymbol{h}_{t,i}^{d,l+2}) \\ \boldsymbol{k}_{s,i}^{e,l+2} = \varphi_{kd}(\boldsymbol{h}_{s,i}^{e,l+2}) \\ \boldsymbol{v}_{s,i}^{e,l+2} = \varphi_{vd}(\boldsymbol{h}_{s,i}^{e,l+2}) \end{array}\right\} \implies \alpha_{ts,i}^{d,l+2} = \texttt{Softmax}\left(\left\langle \boldsymbol{q}_{t,i}^{d,l+2}, \boldsymbol{k}_{t,i}^{e,l+2}\right\rangle\right), \tag{7}$$

$$\boldsymbol{h}_{t,i}^{d,l+3} = \boldsymbol{h}_{t,i}^{d,l+2} + \varphi_{ht}\left(\boldsymbol{h}_{t,i}^{d,l+2}, \sum_{s=1}^{t} \alpha_{ts,i}^{d,l+2}\boldsymbol{v}_{s,i}^{e,l+2}\right),$$

$$\vec{\boldsymbol{x}}_{t,i}^{d,l+3} = \vec{\boldsymbol{x}}_{t,i}^{d,l+2} + \sum_{s=1}^{t} \alpha_{ts,i}^{d,l+2} \cdot \varphi_{xt}\left(\boldsymbol{v}_{s,i}^{e,l+2}\right) \cdot \left(\vec{\boldsymbol{x}}_{t,i}^{d,l+2} - \vec{\boldsymbol{x}}_{s,i}^{e,l+2}\right). \tag{8}$$

Note that when performing the cross-attention between $\mathcal{G}_\delta^{d,l+2}$ and $\mathcal{G}_s^{e,l+2}$, we will first accomplish mean reduction for the coordinates in $\mathcal{G}_s^{e,l+2}$.

### (3) Long-Term MSE Loss

We denote the output of the decoder as $\vec{\boldsymbol{x}}_{T+r-1,i}'$ for the last frame and $\vec{\boldsymbol{x}}_{\delta,i}'$ for the TDG. The prediction of the next frame is given by $\vec{\boldsymbol{x}}_{T+r,i}' = \vec{\boldsymbol{x}}_{T+r-1,i}' + \vec{\boldsymbol{x}}_{\delta,i}'$. This predicted frame is then concatenated into the original sequence, forming a new input for the model, and this process continues until we predict the final frame $\vec{\boldsymbol{x}}_{T+R}'$.

We collect all the long-term predictions and employ Mean Squared Error (MSE) to compute prediction errors, which can be formalized as:

$$\mathcal{L} = \sum_{r=1}^{R}\sum_{i=1}^{N} \texttt{MSE}(\vec{\boldsymbol{x}}_{T+r,i}, \vec{\boldsymbol{x}}_{T+r,i}'). \tag{9}$$

Table 1: Predicted MSE ($\times 10^{-2}$) on MD17 dataset with 10 rollout steps. Results averaged across three trials.

| | Aspirin | Benzene | Ethanol | Malonaldehyde | Naphthalene | Salicylic | Toluene | Uracil |
|---|---|---|---|---|---|---|---|---|
| ST_GNN | $9.403_{\pm 0.150}$ | $1.942_{\pm 0.086}$ | $2.650_{\pm 0.001}$ | $7.203_{\pm 0.102}$ | $4.311_{\pm 0.172}$ | $5.565_{\pm 0.251}$ | $4.530_{\pm 0.061}$ | $4.028_{\pm 0.374}$ |
| ST_TFN | $7.974_{\pm 0.025}$ | $2.084_{\pm 0.001}$ | $2.441_{\pm 0.001}$ | $6.228_{\pm 0.066}$ | $4.768_{\pm 0.078}$ | $6.737_{\pm 0.024}$ | $4.041_{\pm 0.198}$ | $5.672_{\pm 0.098}$ |
| STGCN | $8.079_{\pm 0.001}$ | $1.993_{\pm 0.004}$ | $2.786_{\pm 0.001}$ | $6.464_{\pm 0.001}$ | $5.829_{\pm 0.001}$ | $6.739_{\pm 0.001}$ | $4.724_{\pm 0.001}$ | $6.119_{\pm 0.001}$ |
| ST_SE(3)-Tr. | $6.943_{\pm 0.082}$ | $2.085_{\pm 0.006}$ | $2.079_{\pm 0.001}$ | $5.775_{\pm 0.016}$ | $4.443_{\pm 0.046}$ | $5.577_{\pm 0.021}$ | $3.292_{\pm 0.004}$ | $4.914_{\pm 0.042}$ |
| ST_EGNN | $7.945_{\pm 0.040}$ | $3.764_{\pm 1.834}$ | $1.385_{\pm 0.001}$ | $4.661_{\pm 0.084}$ | $4.226_{\pm 0.752}$ | $6.214_{\pm 0.232}$ | $3.405_{\pm 0.178}$ | $3.303_{\pm 0.291}$ |
| AGL-STAN | $11.885_{\pm 0.697}$ | $5.813_{\pm 0.278}$ | $3.052_{\pm 0.286}$ | $21.715_{\pm 0.617}$ | $2.248_{\pm 0.198}$ | $4.871_{\pm 0.928}$ | $1.909_{\pm 0.053}$ | $1.697_{\pm 0.329}$ |
| EqMotion | $8.433_{\pm 0.001}$ | $4.724_{\pm 0.001}$ | $4.275_{\pm 0.001}$ | $6.787_{\pm 0.001}$ | $6.538_{\pm 0.001}$ | $7.227_{\pm 0.001}$ | $4.922_{\pm 0.001}$ | $6.369_{\pm 0.001}$ |
| ESTAG | $2.553_{\pm 0.414}$ | $1.524_{\pm 0.142}$ | $0.977_{\pm 0.001}$ | $2.758_{\pm 0.794}$ | $2.278_{\pm 0.211}$ | $2.239_{\pm 0.576}$ | $1.733_{\pm 0.591}$ | $1.600_{\pm 0.237}$ |
| GST-F | $\underline{2.345}_{\pm 0.077}$ | $\underline{0.873}_{\pm 0.098}$ | $\underline{0.968}_{\pm 0.001}$ | $\mathbf{1.442}_{\pm 0.025}$ | $\underline{1.297}_{\pm 0.185}$ | $\underline{1.895}_{\pm 0.034}$ | $\mathbf{0.957}_{\pm 0.117}$ | $\underline{1.470}_{\pm 0.234}$ |
| GST-V | $\mathbf{2.196}_{\pm 0.075}$ | $\mathbf{0.480}_{\pm 0.050}$ | $\mathbf{0.940}_{\pm 0.001}$ | $\underline{1.762}_{\pm 0.054}$ | $\mathbf{0.988}_{\pm 0.016}$ | $\mathbf{1.733}_{\pm 0.031}$ | $\underline{1.002}_{\pm 0.063}$ | $\mathbf{1.087}_{\pm 0.055}$ |

To mitigate cumulative errors during the autoregressive inference, we implement a teacher-forcing strategy in the training phase, implying that $\vec{x}'_{T+r,i}$ is estimated through the input of the ground-truth sequence rather than the predicted one. It is noteworthy that all baselines in this paper adhere to same strategies during both training and testing phases, ensuring a fair analysis. A fundamental characteristic of our GST is its E(3)-equivariance property, and the corresponding proof is provided in Appendix A.

## 4 EXPERIMENTS

In this section, we evaluate the performance of GST on long-term prediction tasks across datasets of varying scales, encompassing molecules (§ 4.1), human motions (§ 4.2), and proteins (§ 4.3). To accelerate the dynamics simulations, we follow the sampling approach utilized in previous works (Huang et al., 2022) to acquire the subset of the trajectories for training, validation and testing. Specifically, we randomly select a starting point and subsequently sample $T + 20$ timestamps. The initial $T$ timestamps serve as input observations for the models, while the subsequent 10, 15, and 20 timestamps are future frames to be predicted, depending on the specific task settings about the rollout steps. In § 4.4, we conduct ablation studies to explore the impact of each component on model performance. Additionally, we perform exploratory experiments to identify potentially optimal encoder-decoder structures. More detailed experimental results are also provided in Appendix C.

**Baselines and Metrics.** We benchmark our method against the following baseline approaches, including GNNs such as ST_GNN (Gilmer et al., 2017), STGCN (Yu et al., 2017), and AGL-STAN (Sun et al., 2022), as well as equivariant GNNs such as ST_TFN (Thomas et al., 2018), ST_SE(3)-Tr. (Fuchs et al., 2020), ST_EGNN (Satorras et al., 2021b), ST_GMN (Huang et al., 2022), Eqmotion (Xu et al., 2023), and ESTAG (Wu et al., 2024). Models prefixed with "ST" indicate that we have adapted them to accommodate multi-frame input by incorporating trivial spatio-temporal aggregation, following the setup in Wu et al. (2024). We also report the performance for two versions of the GST: GST-V and GST-F. GST-V (GST-Variable) receives variable-length historical frames as input, while GST-F (GST-Fixed) adopts the same configuration as other baselines, accepting only fixed-length historical frame inputs. As per Eq. (9), we employ the sum of MSEs across all predicted frames during the rollout process as our evaluation metric.

### 4.1 MOLECULAR DYNAMICS

**Dataset and Implementation.** MD17 comprises dynamic trajectories generated by MD simulations for eight distinct small molecules (*e.g.*, aspirin, benzene, etc). We utilize atomic numbers as invariant input features $\boldsymbol{h}_{t,i}$. Further details, including hyper-parameters are provided in Appendix B.1.

**Results.** Table 1 presents the performance of all models in MD17, with a rollout step of 10. The following insights can be drawn from this table: **1.** Our model achieves state-of-the-art performance across all eight small molecules, demonstrating the superiority of our approach in long-term au-

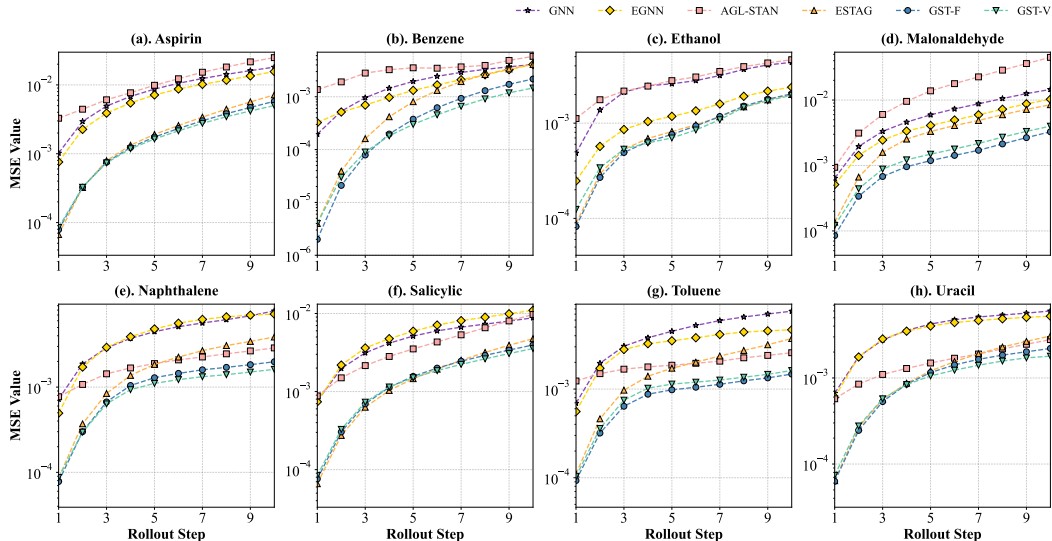

Figure 3: MSEs on MD17 up to 10 rollout steps (better zoom in).

toregressive prediction tasks. **2.** GST-V outperforms GST-F on most molecules, underscoring the capability of our proposed architecture to flexibly process and leverage extended historical information, thereby enhancing prediction accuracy. **3.** Our method outperforms the previous state-of-the-art method ESTAG, seven of the eight small molecules studied, with ethanol as the exception. We posit that may be attributed to ethanol's relatively simple structure, containing only three heavy atoms (two carbon and one oxygen), may result in less discernible performance differences among modeling approaches.

To provide a more intuitive understanding of the prediction accuracy across different models at each step of the rollout process, we present Fig. 3, which illustrates the MSE for various models over a 10-step rollout. It clearly demonstrates that while our model may not achieve optimal performance in the initial steps for some molecules, its advantages in mitigating long-range cumulative errors become increasingly evident as the rollout progresses.

We also evaluate performance on MD17 using extended rollout steps of 15 and 20, allowing us to assess longer-term predictions more effectively. Besides, we modify most baseline models to accept full historical inputs by employing the temporal self-attention mechanism similar to our GNS, except for AGL-STAN and Eqmotion due to architectural constraints. Detailed results and corresponding analyses are presented in Tables 6 to 8 of Appendix C. GST consistently outperforms state-of-the-art methods across almost all molecules, even under these more challenging conditions. Notably, the performance gap widens for several molecules as rollout steps increase, providing strong evidence of GST's efficacy in mitigating cumulative errors and enhancing long-term simulation accuracy.

## 4.2 MOTION CAPTURE

**Dataset and Implementation.** We evaluate our model's performance across various scenarios depicting 3D trajectories of human motion. We focus on two motions: Subject #35 (`Walk`) and Subject #102 (`Basketball`). The more details, including adaptive modifications to other baselines and hyper-parameter settings, are presented in Appendix B.2.

**Results.** As illustrated in Table 2, our model achieves the best perofrmance across different long-term rollout prediction tasks on the Motion Capture dataset. In this table, we denote predicted MSE values exceeding 1000 with a dash (-) symbol. Additionally, we omit the results for Eqmotion, as its predicted MSE values surpassed 1000 across all the settings. It is noteworthy that due to the complexity and rapid changes inherent in motion capture, the predicted MSE for all models increases rapidly as the rollout steps extends. Correspondingly, the performance gap between our method and other baselines widens as the number of rollout steps increases. Furthermore, Fig. 4 showcases the visualization of the predicted motions in `Basketball` and `Walk` by various methods.

Table 2: Predicted MSE ($\times 10^{-1}$) on Motion dataset with 10 rollout steps. We denote predicted MSE values exceeding 1000 with a dash (-) symbol. Results averaged across three trials.

| Method | Walk | | | Basketball | | |
|---|---|---|---|---|---|---|
| | R=10 | R=15 | R=20 | R=10 | R=15 | R=20 |
| ST_GNN | $18.560_{\pm 1.215}$ | $55.062_{\pm 8.120}$ | $97.593_{\pm 9.933}$ | - | - | - |
| ST_TFN | $19.689_{\pm 0.631}$ | $99.021_{\pm 28.136}$ | $201.075_{\pm 11.1}$ | $178.689_{\pm 2.477}$ | $593.498_{\pm 32.395}$ | - |
| ST_GCN | $1.870_{\pm 0.001}$ | $7.418_{\pm 0.001}$ | $13.899_{\pm 0.001}$ | $87.185_{\pm 0.001}$ | $312.096_{\pm 0.001}$ | $531.535_{\pm 0.001}$ |
| ST_SE(3)-Tr. | $8.196_{\pm 1.000}$ | $44.096_{\pm 7.873}$ | $164.483_{\pm 9.169}$ | $183.933_{\pm 6.797}$ | $580.178_{\pm 12.324}$ | - |
| ST_EGNN | $35.863_{\pm 3.156}$ | - | - | - | - | - |
| AGL-STAN | $42.409_{\pm 0.001}$ | $64.652_{\pm 0.001}$ | $170.848_{\pm 0.001}$ | - | - | - |
| ESTAG | $1.418_{\pm 0.087}$ | $5.907_{\pm 0.885}$ | $17.431_{\pm 3.424}$ | $10.209_{\pm 0.071}$ | $54.513_{\pm 1.682}$ | $175.950_{\pm 3.073}$ |
| GST-F | $\mathbf{0.931}_{\pm 0.097}$ | $\underline{4.192}_{\pm 0.765}$ | $\mathbf{11.439}_{\pm 2.408}$ | $\underline{9.712}_{\pm 0.203}$ | $\underline{49.754}_{\pm 0.587}$ | $\underline{155.296}_{\pm 9.479}$ |
| GST-V | $\underline{1.095}_{\pm 0.142}$ | $\mathbf{4.084}_{\pm 0.311}$ | $\underline{12.708}_{\pm 1.466}$ | $\mathbf{9.658}_{\pm 0.072}$ | $\mathbf{49.374}_{\pm 1.410}$ | $\mathbf{148.988}_{\pm 6.631}$ |

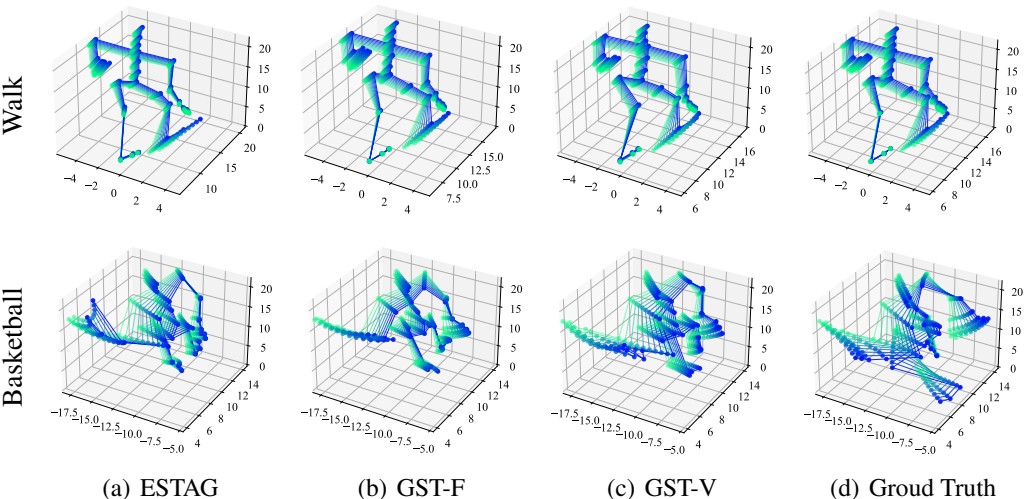

| (a) ESTAG | (b) GST-F | (c) GST-V | (d) Ground Truth |

Figure 4: Visualizations of predicted human motions: `Walk` (top) and `Basketball` (bottom).

## 4.3 PROTEIN DYNAMICS

**Dataset and Implementation.** We leveraged the MDAnalysis toolkit (Gowers et al., 2019) to facilitate the exploration of model's dynamics simulation capabilities on the Adk equilibrium trajectory dataset (Seyler & Beckstein, 2017). To mitigate the computational burden associated with the large number of atoms in protein data, we represented each residue solely by its backbone atoms. The more details, including adaptive modifications to other baselines and hyper-parameter settings, are presented in Appendix B.3.

**Results.** Table 3 illustrates the performance of all models on the protein dataset. Our GST consistently achieves superior performance across various long-term rollout prediction tasks with rollout steps of 10, 15, and 20. This demonstrates that even in systems with complex structures such as proteins, GST can still efficiently extract and utilize critical information from historical timestamps, enabling more accurate long-range predictions.

Table 3: Predicted MSE ($\times 10^{-2}$) on Protein dataset with 10 rollout steps. Results averaged across three trials. Standard deviations are omitted due to their negligible magnitude.

| Method | R=10 | R=15 | R=20 |
|---|---|---|---|
| ST_GNN | 2.196 | 3.108 | $\underline{4.202}$ |
| ST_GCN | 2.285 | 3.700 | 5.733 |
| ST_EGNN | $\underline{2.000}$ | $\underline{3.051}$ | 4.239 |
| AGL-STAN | 2.216 | 3.373 | 4.309 |
| ST_GMN | 2.006 | 3.056 | 4.247 |
| ESTAG | 2.009 | 3.065 | 4.259 |
| GST-F | 2.008 | 3.063 | 4.258 |
| GST-V | **1.911** | **2.971** | **4.048** |

Table 4: Ablation studies ($\times 10^{-3}$) on MD17 dataset with 10 rollout steps.

|                   | Aspirin | Benzene | Ethanol | Malonaldehyde | Naphthalene | Salicylic | Toluene | Uracil |
|-------------------|---------|---------|---------|---------------|-------------|-----------|---------|--------|
| w/o Equivariance  | 30.299  | 2.660   | 21.896  | 24.474        | 35.048      | 31.365    | 32.874  | 31.252 |
| w/o TDG           | 7.849   | 1.150   | 1.991   | 5.321         | 7.000       | 7.227     | 3.301   | 4.379  |
| Only TDG          | 2.714   | 1.303   | 1.167   | 2.143         | 1.480       | 1.940     | 1.262   | 1.121  |
| Decoder-only      | **1.680** | 0.545 | 1.104   | 2.071         | 1.261       | 1.929     | 1.420   | 1.327  |
| Full-Attention    | 2.206   | 0.543   | 0.980   | 2.009         | **0.934**   | 1.889     | 1.035   | 1.095  |
| Shared Parameters | 1.757   | 0.665   | 1.018   | 1.902         | 1.478       | **1.548** | 1.284   | 1.172  |
| GST-V             | 2.195   | **0.480** | **0.940** | **1.761**   | 0.988       | 1.733     | **1.002** | **1.087** |

## 4.4 ABLATION STUDIES

To validate the contribution of each module to our GST's overall performance and to explore the intricacies of the Encoder-Decoder framework in physical dynamics scenarios, we conducted ablation and exploratory experiments. The results are presented in Table 4.

Our observations and findings are as follows: **1.** Disregarding equivariance led to a significant performance degradation across all molecules (Row 1). This underscores the critical role of physical symmetry in modeling 3D physical dynamics. **2.** To assess the impact of the TDG, we remove the TDG from the decoder (Row 2). The results indicate that the absence of TDG substantially impairs model performance, suggesting a strong correlation between physical dynamics trajectories and local frames of the input sequence, particularly the final two frames. **3.** We experiment with modifying the decoder input to solely include the TDG (Row 3). This modification resultes in a notable performance decline, indicating the necessity of allowing TDG and original sequence to interact, thereby extracting crucial temporal information. **4.** To evaluate the benefits of the encoder-decoder structure, we implement a decoder-only structurte (Row 4). The results demonstrate that the encoder-decoder architecture outperforms the decoder-only variant for the majority of molecules. **5.** We adopt the masking strategy from standard Transformer, replacing the current causal-attention in the encoder's self-attention module with full-attention (Row 5). This substitution led to a slight performance decrease for most molecules, suggesting that when performing self-attention on temporal data, the model must adhere to objective physical laws, allowing the current frame to be updated based solely on previous frames. **6.** To ascertain whether the performance improvement is due to the increased parameters introduced by the encoder-decoder architecture, we share parameters between the encoder and decoder modules, effectively halving the total parameters of the model (Row 6). We observe performance decreases for some molecules, but improvements for others. Notably, when computing the average across all eight molecules, the shared-parameter version exhibited only a marginal difference in performance compared to the original version. This further emphasizes the potential of the encoder-decoder architecture in enhancing the model's long-term prediction capabilities.

## 5 CONCLUSION

In this paper, we introduce Geometric Spatiotemporal Transformers (GST), a novel framework that integrates the original Transformer architecture with physical symmetries to simulate long-term physical dynamics. Leveraging the inherent characteristics of the Transformer, GST efficiently processes historical inputs of arbitrary length. The overall GST architecture ensures E(3)-equivariance, effectively encoding physical symmetries and enhancing the modeling capability for 3D objects. The incorporation of a Temporal Difference Graph (TDG) significantly mitigates cumulative errors generated during long-term rollouts process. Extensive experiments across various scales (molecules, proteins, and human motions) and diverse rollout steps demonstrate GST's superior performance in physical dynamics simulation. We envision GST serving as a robust baseline in this field, with broad applicability across various tasks, including drug discovery, robot control, and materials design.

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

## A  EQUIVARIANCE OF GST

We know that the $E(3)$ symmetry can be decomposed into the symmetry on the three-dimensional translation group $T(3)$ and the symmetry on the three-dimensional orthogonal group $O(3)$. We will prove these two separately below.

### A.1  TRANSLATION EQUIVARIANCE OF GST

With mean reduction (Puny et al., 2021), translational variability can always be easily achieved. We just need to prove that subsequent operations are translation invariant. In fact, there are only Eqs. (2) and (4) to (6) require to prove its translation invariant. And we find that, all sums of the coefficients of these coordinate terms is $1 + (-1) = 0$, which indicates its translation invariance.

### A.2  $O(3)$-EQUIVARIANCE OF GST

$O(3)$**-equivariance of whole model.** Since the equivariance can be understood that group actions can be exchanged with mappings of any layer, so the equivariance of the entire model can be proved by proving that the individual modules of the model are equivariant. In general, we only need to prove the following three points:

1. $O(3)$-equivariance of spatial module.

2. $O(3)$-equivariance of temporal module.

3. $O(3)$-invariance of objective function.

Here we give the proof of $O(3)$-equivariance by the symbology from e3nn (Geiger & Smidt, 2022). $O(3)$ consists of rotation and inverse, implying $O(3) = SO(3) \times C_i$, where $SO(3)$ is the rotation group and $C_i = \{\mathfrak{e}, \mathfrak{i}\}$ denotes the inverse group. We thus specify the group representation of $O(3)$ as

$$\rho^{(l)}(\mathfrak{r}\mathfrak{m}) := \sigma^{(l,p)}(\mathfrak{m})\boldsymbol{D}^{(l)}(\mathfrak{r}), \tag{10}$$

where $p \in \{\pm 1\}$ called *parity*. $\sigma^{(l)}(\mathfrak{m}) = 1$ for $\mathfrak{m} = \mathfrak{e}$ (the identity) , $\sigma^{(l)}(\mathfrak{m}) = p^l$ if $\mathfrak{m} = \mathfrak{i}$ (the inverse), and $\boldsymbol{D}^{(l)}(\mathfrak{r})$ is the Wigner-D matrix of $l$-degree. With Eq. (10) of $l$ be the representation, features be the solution of such equivariant constraint equation is call $l$-degree steerable features. Moreover, we donate a $l$-degree vector as $l\mathtt{o}$ if $p = -1$ (inverse equivariant), and $l\mathtt{e}$ if $p = 1$ (inverse invariant).

**Example A.1** (Types of common equivariant/invariant features)**.** Considering that the highest degree of features in this article does not exceed 1, we only introduce four common cases: $0\mathtt{e}$, $0\mathtt{o}$, $1\mathtt{o}$, and $1\mathtt{e}$.

- Scalar (*e.g.* charges, distance and potential energy) is invariant to both rotation and inverse, thus denoted by $0\mathtt{e}$.

- Pseudo-scalar (*e.g.* triple product of three 3-dimensional vectors, magnetic flux and helicity) is invariant to rotation but equivariant to inverse, thus denoted by $0\mathtt{o}$.

- Vector (*e.g.* postion, velocity and acceleration) is equivariant to both rotation and inverse, thus denoted by $1\mathtt{o}$.

- Pseudo-vector (*e.g.* angular momentum, torque and magnetic field vector) is equivariant to rotation but invariant to inverse, thus denoted by $1\mathtt{e}$.

In the scope of consideration of this article, there are only two types of operations to change the degree of features: $(0\mathtt{e})\cdot(1\mathtt{o}) \to (1\mathtt{o})$ and transforming $(1\mathtt{o})$ into $(0\mathtt{e})$ through norm $\|1\mathtt{o}\| \to 0\mathtt{e}$. From this perspective, proving the equivariance of a model requires only pointing out all the types of variables. And it is worth noting that all inputs inside the function are invariants, which is denoted as $[0\mathtt{e}, 0\mathtt{e}, \dots, 0\mathtt{e}] \to 0\mathtt{e}$.

$O(3)$**-equivariance of spatial module.** To prove the spatial module is $O(3)$-equivariant, we only require to prove message $\boldsymbol{m}_{t,ij}^{e,l}$ and updated node feature $\boldsymbol{h}_{t,i}^{e,l+1}$ are $0\mathtt{e}$-features and updated node

positions $\vec{\boldsymbol{x}}_{t,i}^{e,l+1}$ are $\texttt{1o}$-features. Based on the symbology, we can easily see this by annotating Eq. (2) follows:

$$
\begin{aligned}
\underbrace{\boldsymbol{m}_{t,ij}^{e,l}}_{\texttt{0e}} &= \underbrace{\varphi_m\left(\boldsymbol{h}_{t,i}^{e,l},\, \boldsymbol{h}_{t,j}^{e,l},\, \left\|\vec{\boldsymbol{x}}_{t,i}^{e,l} - \vec{\boldsymbol{x}}_{t,j}^{e,l}\right\|\right)}_{[\texttt{0e},\, \texttt{0e},\, \|\texttt{1o}\| \to \texttt{0e}] \to \texttt{0e}}, \\
\underbrace{\boldsymbol{h}_{t,i}^{e,l+1}}_{\texttt{0e}} &= \underbrace{\boldsymbol{h}_{t,i}^{e,l}}_{\texttt{0e}} + \underbrace{\varphi_h\left(\boldsymbol{h}_{t,i}^{e,l},\, \sum_{j\in\mathcal{N}(i)} \boldsymbol{m}_{t,j}^{e,l}\right)}_{[\texttt{0e},\, \texttt{0e}] \to \texttt{0e}}, \\
\underbrace{\vec{\boldsymbol{x}}_{t,i}^{e,l+1}}_{\texttt{1o}} &= \underbrace{\vec{\boldsymbol{x}}_{t,i}^{e,l}}_{\texttt{1o}} + \frac{1}{|\mathcal{N}(i)|}\sum_{j\in\mathcal{N}(i)} \underbrace{\varphi_x\left(\boldsymbol{m}_{t,ij}^{e,l}\right)\cdot\left(\vec{\boldsymbol{x}}_{t,i}^{e,l} - \vec{\boldsymbol{x}}_{t,j}^{e,l}\right)}_{\texttt{0e}\,\cdot\,\texttt{1o} \to \texttt{1o}},
\end{aligned}
\tag{11}
$$

$O(3)$**-equivariance of temporal module.** In fact, like most equivariant Graph Transformer models except $SE(3)$-Transformer, the attention mechanism obtains $\texttt{0e}$-features as follows:

$$
\left.
\begin{aligned}
\underbrace{\boldsymbol{q}_{t,i}^{e,l+1}}_{\texttt{0e}} &= \underbrace{\varphi_{qe}(\boldsymbol{h}_{t,i}^{e,l+1})}_{[\texttt{0e}]\to\texttt{0e}} \\
\underbrace{\boldsymbol{k}_{s,i}^{e,l+1}}_{\texttt{0e}} &= \underbrace{\varphi_{ke}(\boldsymbol{h}_{s,i}^{e,l+1})}_{[\texttt{0e}]\to\texttt{0e}} \\
\underbrace{\boldsymbol{v}_{s,i}^{e,l+1}}_{\texttt{0e}} &= \underbrace{\varphi_{ve}(\boldsymbol{h}_{s,i}^{e,l+1})}_{[\texttt{0e}]\to\texttt{0e}}
\end{aligned}
\right\}
\implies
\underbrace{\alpha_{ts,i}^{e,l+1}}_{\texttt{0e}} = \underbrace{\texttt{Softmax}\left(\left\langle \boldsymbol{q}_{t,i}^{e,l+1},\, \boldsymbol{k}_{s,i}^{e,l+1}\right\rangle\right)}_{[\texttt{0e},\, \texttt{0e}]\to\texttt{0e}},
\tag{12}
$$

After that, we only need to prove that updated node features $\boldsymbol{h}_{t,i}^{e,l+1}$ are $\texttt{0e}$-features and updated node positions $\vec{\boldsymbol{x}}_{t,i}^{e,l+1}$ are $\texttt{1o}$-features, just as we did for the $O(3)$-equivariance of spatial module. The details are as follows:

$$
\begin{aligned}
\underbrace{\boldsymbol{h}_{t,i}^{e,l+2}}_{\texttt{0e}} &= \underbrace{\boldsymbol{h}_{t,i}^{e,l+1}}_{\texttt{0e}} + \underbrace{\varphi_{ht}\left(\boldsymbol{h}_{t,i}^{e,l+1},\, \sum_{s=1}^{t}\alpha_{ts,i}^{e,l+1}\boldsymbol{v}_{s,i}^{e,l+1}\right)}_{[\texttt{0e},\texttt{0e}]\to\texttt{0e}}, \\
\underbrace{\vec{\boldsymbol{x}}_{t,i}^{e,l+2}}_{\texttt{1o}} &= \underbrace{\vec{\boldsymbol{x}}_{t,i}^{e,l+1}}_{\texttt{1o}} + \sum_{s=1}^{t}\underbrace{\alpha_{ts,i}^{e,l+1}\varphi_{xt}\left(\boldsymbol{v}_{s,i}^{e,l+1}\right)\cdot\left(\vec{\boldsymbol{x}}_{t,i}^{e,l+1} - \vec{\boldsymbol{x}}_{s,i}^{e,l+1}\right)}_{\texttt{0e}\cdot\texttt{0e}\cdot\texttt{1o}\to\texttt{1o}}.
\end{aligned}
\tag{13}
$$

Note that the formula form may be slightly different when initialized, but we only need to discuss the difference about translation equivariance/invariance, and we will not repeat the proof of rotation/inverse equivariance/invariance here.

$O(3)$**-invariance of objective function.** Note that $\texttt{MSE}(\cdot)$ is a simple modulo length operator, so its $O(3)$-invariance is obvious.

$$
\underbrace{\mathcal{L}}_{\texttt{0e}} = \sum_{r=1}^{R}\sum_{i=1}^{N}\texttt{MSE}(\vec{\boldsymbol{x}}_{T+r,i}^{*},\, \vec{\boldsymbol{x}}_{T+r,i}) = \sum_{r=1}^{R}\sum_{i=1}^{N}\underbrace{\left\|\vec{\boldsymbol{x}}_{T+r,i}^{*} - \vec{\boldsymbol{x}}_{T+r,i}\right\|}_{\|\texttt{1o}\|\to\texttt{0e}},
\tag{14}
$$

# B    IMPLEMENTATION DETAILS

## B.1    IMPLEMENTATION DETAILS ON MD17 DATASET

The first column of Table 5 presents a unified set of hyper-parameters employed consistently across all experimental evaluations on the MD17 datasets. These parameters are uniformly applied to both our proposed GST model and all baseline methods. Both our GST and all other baselines are trained and tested on a single NVIDIA A100-80G GPU. The number of training, validation and testing sets are 500, 2000 and 2000, respectively.

In the MD17 dataset, each temporal graph contains up to 13 nodes. As for graph construction, We compute pairwise distances between all atoms, designating those within a distance threshold $\lambda$

Table 5: Hyper-parameters of GST and other baseline methods. The temporal length $T$ denotes the length of initial timestamps, the time lag $\Delta t$ denotes the interval between two timestamps, the hidden size denotes the size of hidden states, and the layer denotes the number of layers.

| Hyper-parameter | MD17 | Protein | Motion Capture |
|---|---|---|---|
| Learning Rate | 5e-3 | 5e-5 | 5e-3 |
| Epochs | 500 | 500 | 500 |
| Temporal Length $T$ | 10 | 10 | 10 |
| Time Lag $\Delta t$ | 10 | 5 | 1 |
| Hidden Size | 16 | 16 | 16 |
| Layer | 2 | 2 | 2 |

as first-order neighbors. Each atom forms edges with both its first- and second-order neighbors to facilitate subsequent message passing.

### B.2 IMPLEMENTATION DETAILS ON MOTION CAPTURE

The third column of Table 5 presents a unified set of hyper-parameters employed consistently across all experimental evaluations on the motion capture datasets. These parameters are uniformly applied to both our proposed GST model and all baseline methods. Both our GST and all other baselines are trained and tested on a single NVIDIA A100-80G GPU. We adopt the setups and data splits from the official code of (Wu et al., 2024). The subject #35 (`Walk`) contains 1100/600/600 trajectories for training/validation/testing, while the subject #102 (`Basketball`) contains 600/300/300 trajectories for training/validation/testing.

In the Motion Capture dataset, each temporal graph contains up to 31 nodes. As for graph construction, directly connected joint nodes are classified as first-order neighbors. Each joint node establishes connections with both its first- and second-order neighbors to facilitate subsequent message passing. The invariant input features $h$ of all the joints are all 1s. In this experimental setup, we set $\Delta t = 1$, considering the substantial positional variations in motion data compared to MD17 and protein datasets. This choice mitigates the exponential growth of cumulative errors in long-term predictions. A larger $\Delta t$ would lead to unacceptably high error magnitudes across all methods as rollout steps increase, rendering the comparisons across different methods infeasible and compromising the validity of our comparative analysis.

### B.3 IMPLEMENTATION DETAILS ON PROTEIN DATASET

The second column of Table 5 presents a unified set of hyper-parameters employed consistently across all experimental evaluations on the protein datasets. These parameters are uniformly applied to both our proposed GST model and all baseline methods. Both our GST and all other baselines are trained and tested on a single NVIDIA A100-80G GPU. The dataset is partitioned into training, validation, and test sets with a ratio of 6:2:2, respectively. This partition resulted in 2,482 samples for training, 827 for validation, and 827 for testing.

In the Protein dataset, each temporal graph contains 5325 nodes. We construct a 4-channel equivariant 3D coordinate for the four backbone atoms (N, $C_\alpha$, C, O) , combining it with the corresponding atomic numbers as invariant input features $h$ to represent a single node in the graph. This node representation methodology is consistently applied across all baseline methods for comparative integrity. As for graph construction, we compute pairwise distances between all atoms, designating those within a distance threshold $\lambda$ as first-order neighbors. Each atom forms edges with its first-order neighbor to facilitate subsequent message passing.

# C   MORE EXPERIMENT RESULTS ON MD17

## C.1   VARIABLE-INPUT FOR OTHER BASELINES

Table 6: Prediction error ($\times 10^{-2}$) on MD17 dataset. Results averaged across 3 runs.

|  | Aspirin | Benzene | Ethanol | Malonaldehyde | Naphthalene | Salicylic | Toluene | Uracil |
|---|---|---|---|---|---|---|---|---|
| ST_GNN-F | $9.403_{\pm 0.150}$ | $1.942_{\pm 0.086}$ | $2.650_{\pm 0.001}$ | $7.203_{\pm 0.102}$ | $4.311_{\pm 0.172}$ | $5.565_{\pm 0.251}$ | $4.530_{\pm 0.061}$ | $4.028_{\pm 0.374}$ |
| ST_GNN-V | $10.879_{\pm 0.001}$ | $2.174_{\pm 0.001}$ | $2.935_{\pm 0.001}$ | $9.134_{\pm 0.126}$ | $5.554_{\pm 0.020}$ | $7.035_{\pm 0.117}$ | $4.520_{\pm 0.140}$ | $4.626_{\pm 0.020}$ |
| ST_TFN-F | $7.974_{\pm 0.025}$ | $2.084_{\pm 0.001}$ | $2.441_{\pm 0.001}$ | $6.228_{\pm 0.066}$ | $4.768_{\pm 0.078}$ | $6.737_{\pm 0.024}$ | $4.041_{\pm 0.198}$ | $5.672_{\pm 0.098}$ |
| ST_TFN-V | $8.614_{\pm 0.027}$ | $2.270_{\pm 0.001}$ | $2.426_{\pm 0.001}$ | $7.247_{\pm 0.351}$ | $4.860_{\pm 0.057}$ | $6.870_{\pm 0.066}$ | $4.032_{\pm 0.143}$ | $5.623_{\pm 0.043}$ |
| STGCN-F | $8.079_{\pm 0.001}$ | $1.993_{\pm 0.004}$ | $2.786_{\pm 0.001}$ | $6.464_{\pm 0.001}$ | $5.829_{\pm 0.001}$ | $6.739_{\pm 0.001}$ | $4.724_{\pm 0.001}$ | $6.119_{\pm 0.001}$ |
| STGCN-V | $8.100_{\pm 0.001}$ | $2.240_{\pm 0.004}$ | $2.785_{\pm 0.001}$ | $6.467_{\pm 0.001}$ | $5.840_{\pm 0.001}$ | $6.976_{\pm 0.001}$ | $4.724_{\pm 0.001}$ | $6.175_{\pm 0.001}$ |
| ST_SE(3)-Tr.-F | $6.943_{\pm 0.082}$ | $2.085_{\pm 0.006}$ | $2.079_{\pm 0.001}$ | $5.775_{\pm 0.016}$ | $4.443_{\pm 0.046}$ | $5.577_{\pm 0.021}$ | $3.292_{\pm 0.004}$ | $4.914_{\pm 0.042}$ |
| ST_SE(3)-Tr.-V | $7.750_{\pm 0.062}$ | $2.232_{\pm 0.001}$ | $2.159_{\pm 0.001}$ | $6.810_{\pm 0.012}$ | $4.515_{\pm 0.019}$ | $6.063_{\pm 0.036}$ | $3.312_{\pm 0.012}$ | $5.185_{\pm 0.038}$ |
| ST_EGNN-F | $7.945_{\pm 0.040}$ | $3.764_{\pm 1.834}$ | $1.385_{\pm 0.001}$ | $4.661_{\pm 0.084}$ | $4.226_{\pm 0.752}$ | $6.214_{\pm 0.232}$ | $3.405_{\pm 0.178}$ | $3.303_{\pm 0.291}$ |
| ST_EGNN-V | $7.350_{\pm 0.589}$ | $1.922_{\pm 0.044}$ | $1.913_{\pm 0.001}$ | $5.183_{\pm 0.008}$ | $3.753_{\pm 0.145}$ | $5.536_{\pm 0.765}$ | $2.812_{\pm 0.109}$ | $3.845_{\pm 0.369}$ |
| ESTAG-F | $2.553_{\pm 0.414}$ | $1.524_{\pm 0.142}$ | $0.977_{\pm 0.001}$ | $2.758_{\pm 0.794}$ | $2.278_{\pm 0.211}$ | $2.239_{\pm 0.576}$ | $1.733_{\pm 0.591}$ | $1.600_{\pm 0.237}$ |
| ESTAG-V | $2.679_{\pm 0.193}$ | $1.699_{\pm 0.331}$ | $1.260_{\pm 0.001}$ | $3.098_{\pm 0.154}$ | $1.556_{\pm 0.126}$ | $2.406_{\pm 0.228}$ | $1.436_{\pm 0.223}$ | $1.805_{\pm 0.030}$ |
| GST-F | $\underline{2.345}_{\pm 0.077}$ | $\underline{0.873}_{\pm 0.098}$ | $\underline{0.968}_{\pm 0.001}$ | $\mathbf{1.442}_{\pm 0.025}$ | $\underline{1.297}_{\pm 0.185}$ | $\underline{1.895}_{\pm 0.034}$ | $\mathbf{0.957}_{\pm 0.117}$ | $\underline{1.470}_{\pm 0.234}$ |
| GST-V | $\mathbf{2.196}_{\pm 0.075}$ | $\mathbf{0.480}_{\pm 0.050}$ | $\mathbf{0.940}_{\pm 0.001}$ | $\underline{1.762}_{\pm 0.054}$ | $\mathbf{0.988}_{\pm 0.016}$ | $\mathbf{1.733}_{\pm 0.031}$ | $\underline{1.002}_{\pm 0.063}$ | $\mathbf{1.087}_{\pm 0.055}$ |

In Table 1 of the main text, all baseline methods employ fixed-length historical frame inputs, potentially leading to unfair comparisons. To address this, we adapt most baseline methods to accept variable-length historical frame inputs, ensuring a more equitable evaluation. However, two methods were left unmodified: AGL-STAN, due to its predefined fixed-size convolutional layers, and Eqmotion, as modifying it would require substantial changes to its initial input, potentially significantly impacting model performance. Consequently, we revise all other baselines accordingly. Table 6 presents the detailed experimental results, where the suffix "F" denotes "fixed-length inputs" version, and "V" represents "variable-length inputs" version.

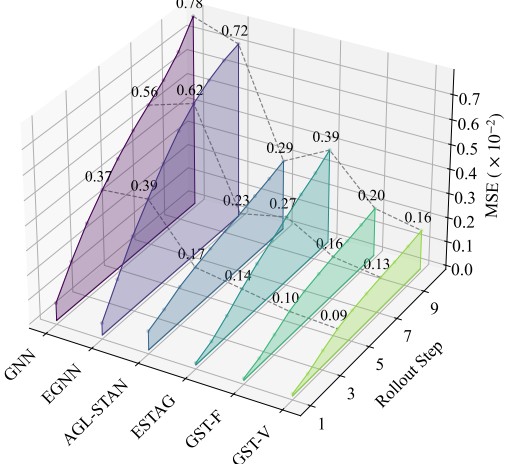

Figure 5: Roll out on Naphthalene (from MD-17).

The experimental results reveal that when models are modified to accept variable-length inputs, ST_EGNN experiences performance degradation on some molecules, while the other four methods show varying degrees of performance decline across most molecules. This observation suggests that these methods struggle to efficiently process and utilize variable-length historical information, highlighting a limitation in their adaptability to more flexible input structures.

To provide a clear and intuitive comparison of different methods' performance on the MD17 dataset, Fig. 5 presents a visualization of the per-step prediction MSE values during long-term rollout for the Naphthalene molecule. This visual representation facilitates a direct assessment of the various approaches' predictive accuracy over extended rollout steps.

## C.2   EXPERIMENT ON MD17 WITH 15 AND 20 ROLLOUT STEPS

In addition to the results presented in the main text for the MD17 dataset with a rollout of 10 steps, we further investigate the model performance with extended rollout lengths of 15 and 20 steps. Tables 7 and 8 provide detailed results for these extended rollout scenarios. Analysis of these tables reveals that our proposed GST model maintains superior performance across the majority of

molecules (8 / 8 for 15 rollout steps and 7 / 8 for 20 rollout steps) as the rollout length increases. Moreover, we observe that the gap in average prediction MSE between GST and other baseline models widens for some molecules as the rollout length increases. These findings provide compelling evidence of GST's excellent performance in long-term physical dynamics simulation tasks, further validating its effectiveness and robustness in challenging predictive scenarios.

Table 7: Predicted MSE ($\times 10^{-2}$) on MD17 dataset with 15 rollout steps. Results averaged across three trials.

| | Aspirin | Benzene | Ethanol | Malonaldehyde | Naphthalene | Salicylic | Toluene | Uracil |
|---|---|---|---|---|---|---|---|---|
| ST_GNN | $21.591_{\pm 0.218}$ | $4.162_{\pm 0.043}$ | $5.127_{\pm 0.001}$ | $17.921_{\pm 0.350}$ | $9.311_{\pm 1.146}$ | $12.922_{\pm 2.272}$ | $8.292_{\pm 0.056}$ | $6.942_{\pm 0.770}$ |
| ST_TFN | $18.264_{\pm 0.050}$ | $4.729_{\pm 0.001}$ | $4.436_{\pm 0.075}$ | $15.429_{\pm 0.301}$ | $9.301_{\pm 0.138}$ | $14.429_{\pm 0.205}$ | $6.528_{\pm 0.112}$ | $11.733_{\pm 0.148}$ |
| STGCN | $18.477_{\pm 0.001}$ | $4.288_{\pm 0.001}$ | $4.785_{\pm 0.001}$ | $15.572_{\pm 0.001}$ | $10.821_{\pm 0.001}$ | $14.065_{\pm 0.001}$ | $8.144_{\pm 0.001}$ | $12.349_{\pm 0.001}$ |
| ST_SE(3)-Tr. | $16.518_{\pm 0.095}$ | $4.726_{\pm 0.020}$ | $3.796_{\pm 0.001}$ | $14.615_{\pm 0.025}$ | $8.321_{\pm 0.041}$ | $12.071_{\pm 0.098}$ | $5.796_{\pm 0.123}$ | $10.286_{\pm 0.086}$ |
| ST_EGNN | $19.298_{\pm 0.551}$ | $85.91_{\pm 60.391}$ | $3.455_{\pm 0.001}$ | $11.984_{\pm 0.737}$ | $10.95_{\pm 1.698}$ | $12.898_{\pm 1.196}$ | $6.061_{\pm 0.442}$ | $6.575_{\pm 0.769}$ |
| AGL-STAN | $31.171_{\pm 1.348}$ | $7.054_{\pm 0.196}$ | $5.834_{\pm 0.194}$ | $61.618_{\pm 3.265}$ | $3.493_{\pm 0.231}$ | $18.208_{\pm 1.420}$ | $3.331_{\pm 0.096}$ | $3.646_{\pm 0.293}$ |
| Eqmotion | $28.426_{\pm 0.001}$ | $41.291_{\pm 0.001}$ | $32.310_{\pm 0.001}$ | $24.564_{\pm 0.001}$ | $18.694_{\pm 0.001}$ | $23.711_{\pm 0.001}$ | $12.649_{\pm 0.001}$ | $17.329_{\pm 0.001}$ |
| ESTAG | $8.801_{\pm 1.030}$ | $3.001_{\pm 0.515}$ | $2.584_{\pm 0.001}$ | $5.648_{\pm 0.771}$ | $2.968_{\pm 1.007}$ | $7.109_{\pm 0.724}$ | $3.473_{\pm 1.153}$ | $4.713_{\pm 0.475}$ |
| GST-F | $\underline{7.071}_{\pm 0.617}$ | $\underline{2.672}_{\pm 0.292}$ | $\mathbf{2.485}_{\pm 0.001}$ | $\underline{5.549}_{\pm 0.126}$ | $\mathbf{2.645}_{\pm 0.204}$ | $\underline{5.468}_{\pm 0.465}$ | $\underline{2.495}_{\pm 0.410}$ | $\underline{3.056}_{\pm 0.839}$ |
| GST-V | $\mathbf{6.881}_{\pm 0.755}$ | $\mathbf{1.905}_{\pm 0.009}$ | $\underline{2.551}_{\pm 0.001}$ | $\mathbf{4.705}_{\pm 0.107}$ | $\underline{2.784}_{\pm 0.432}$ | $\mathbf{5.162}_{\pm 0.220}$ | $\mathbf{2.166}_{\pm 0.251}$ | $\mathbf{2.419}_{\pm 0.208}$ |

Table 8: Predicted MSE ($\times 10^{-2}$) on MD17 dataset with 20 rollout steps. Results averaged across three trials.

| | Aspirin | Benzene | Ethanol | Malonaldehyde | Naphthalene | Salicylic | Toluene | Uracil |
|---|---|---|---|---|---|---|---|---|
| ST_GNN | $39.298_{\pm 1.003}$ | $\underline{6.909}_{\pm 0.310}$ | $8.101_{\pm 0.001}$ | $35.272_{\pm 0.565}$ | $24.729_{\pm 4.049}$ | $18.066_{\pm 1.511}$ | $13.540_{\pm 0.708}$ | $10.506_{\pm 0.976}$ |
| ST_TFN | $33.855_{\pm 0.097}$ | $8.378_{\pm 0.021}$ | $6.561_{\pm 0.023}$ | $30.603_{\pm 0.098}$ | $13.525_{\pm 0.285}$ | $24.963_{\pm 0.051}$ | $10.024_{\pm 1.426}$ | $20.743_{\pm 0.186}$ |
| STGCN | $33.887_{\pm 0.001}$ | $6.391_{\pm 0.001}$ | $6.993_{\pm 0.001}$ | $30.293_{\pm 0.001}$ | $17.026_{\pm 0.001}$ | $23.247_{\pm 0.001}$ | $11.709_{\pm 0.001}$ | $20.991_{\pm 0.001}$ |
| ST_SE(3)-Tr. | $30.851_{\pm 0.093}$ | $8.313_{\pm 0.011}$ | $5.862_{\pm 0.001}$ | $28.374_{\pm 0.047}$ | $13.070_{\pm 0.145}$ | $20.501_{\pm 0.327}$ | $13.673_{\pm 7.056}$ | $17.507_{\pm 0.058}$ |
| ST_EGNN | $46.452_{\pm 5.563}$ | $39.574_{\pm 7.444}$ | $5.225_{\pm 0.001}$ | $23.438_{\pm 0.590}$ | $14.064_{\pm 0.240}$ | $23.724_{\pm 1.444}$ | $9.006_{\pm 0.962}$ | $10.560_{\pm 1.665}$ |
| AGL-STAN | $61.723_{\pm 9.104}$ | $22.043_{\pm 3.660}$ | $10.389_{\pm 0.686}$ | $158.11_{\pm 1.651}$ | $8.398_{\pm 1.199}$ | $23.458_{\pm 3.255}$ | $4.947_{\pm 0.289}$ | $8.064_{\pm 2.291}$ |
| Eqmotion | $191.846_{\pm 0.001}$ | $801.551_{\pm 0.001}$ | $836.737_{\pm 0.001}$ | $103.339_{\pm 0.001}$ | $87.615_{\pm 0.001}$ | $113.888_{\pm 0.001}$ | $135.910_{\pm 0.001}$ | $144.932_{\pm 0.001}$ |
| ESTAG | $21.068_{\pm 1.472}$ | $7.377_{\pm 1.410}$ | $\mathbf{4.317}_{\pm 0.001}$ | $13.538_{\pm 2.102}$ | $7.124_{\pm 1.962}$ | $12.668_{\pm 2.574}$ | $5.270_{\pm 0.614}$ | $7.323_{\pm 0.635}$ |
| GST-F | $\mathbf{16.527}_{\pm 0.101}$ | $7.287_{\pm 0.456}$ | $\underline{4.582}_{\pm 0.001}$ | $\underline{11.631}_{\pm 1.080}$ | $\underline{6.823}_{\pm 2.132}$ | $\underline{11.867}_{\pm 3.392}$ | $\mathbf{3.050}_{\pm 0.428}$ | $\underline{4.963}_{\pm 0.272}$ |
| GST-V | $\underline{17.405}_{\pm 1.891}$ | $\mathbf{5.376}_{\pm 0.501}$ | $5.028_{\pm 0.001}$ | $\mathbf{11.448}_{\pm 1.120}$ | $\mathbf{5.617}_{\pm 2.350}$ | $10.372_{\pm 0.511}$ | $\underline{3.288}_{\pm 0.130}$ | $\mathbf{4.588}_{\pm 0.794}$ |

# D  VISUALIZATION OF THE ATTENTION MAP

In Fig. 6, we present the visualization of the attention map for GST-V. The visualization shows that most timesteps attend to a significant portion of the historical frames. This further validates that GST-V effectively captures and integrates historical input information to enhance prediction performance.

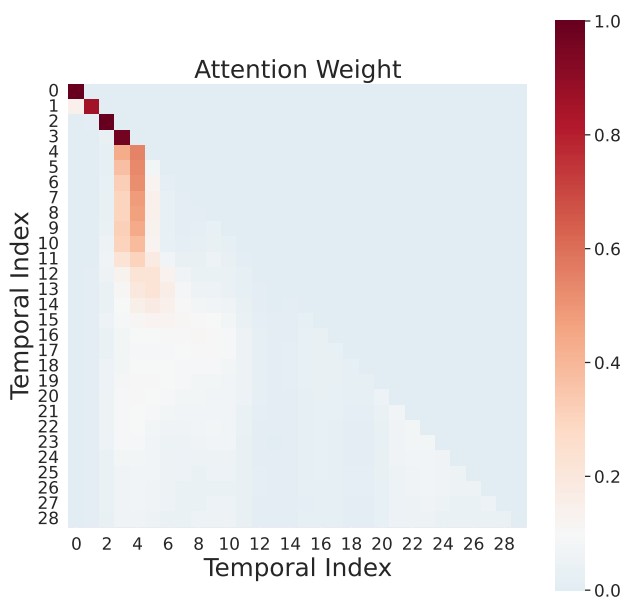

Figure 6: Visualization of the attention map.

