# OpenReview forum: "Geometric Spatiotemporal Transformer to Simulate Long-Term Physical Dynamics"
_ICLR.cc/2025/Conference — Submitted to ICLR 2025_

### Official Review · Reviewer_YGWJ · 2024-10-16

**Soundness:** 3
**Presentation:** 3
**Contribution:** 3
**Rating:** 3
**Confidence:** 4

**Summary:**

The paper proposes multiple components aimed at improving modeling temporal physics simulations with particular focus on long-horizon rollout performance. At the core of the paper is the Temporal Difference Graph (TDG) which aims to mitigate error accumulation during the rollout by using a history of previous frames together with predicting the delta to the next frame instead of the next frame directly. This is combined with equivariant spatiotemporal transformer blocks as core building block. The proposed method shows improvements over state-of-the-art in small-scale molecular dynamics, motion capture and protein dynamics.

**Strengths:**

- Predicting the delta to the next timeframe is a compelling strategy to mitigate error accumulation. This strategy is also commonly used in diffusion models and the paper shows that this strategy also works well on physical simulation rollouts.
- Using transformers instead of GNNs makes a lot of sense as their scalability is well established and their flexibility (e.g. like using the TDG to incorporate arbitrary many history timesteps) is desireable
- The method shows strong improvements over competitors.
- Clear mathematical and visual description of the method.

**Weaknesses:**

- The paper lacks runtime/flops and parameter count comparisons. Even though the architecture and training procedure can be quite different between methods, a comparison of parameter counts and runtimes is essential to exclude performance gains simply due to investing more compute/parameters into the model/training.
- Processing the history (especially in the GST-V) could become quite costly, so the runtime necessary for a rollout should be reported.
- The considered models are extremely small (hidden dim 16 with 2 layers according to the Appendix). This limits the interaction capabilities of GNNs and therefore plays into the hand of transformers, which have global interactions in all layers. A comparison of deeper models or a justification why it is not limiting GNNs to be local should be provided (not sure how many message passing steps are common in these setups).
- The problem sizes should be stated in the paper/appendix, it is only stated how many samples are in a dataset, but not how many nodes a single sample has.
- Autoregressive models are commonly trained with next-step prediction. However due to the time aggregation and history components of the proposed method, a multi-step prediction during training is necessary which greatly limits scalability to larger models, longer rollouts and larger problem sizes (number of nodes)
- The considered datasets are at most 20 rollout timesteps, "long-horizon" seems a bit of a stretch for that.
- Given the similar performance of GST-F and GST-V, it would be interesting to visualize the attention maps to see if GST-V actually uses all of the history or only the recent ones.

**Questions:**

- How do models compare in terms of runtimes/parameters/flops?
- Why are the models so small, are the datasets so small/simple that larger ones run into overfitting? Are larger models infeasible? You are using A100 cards so larger models should not be a problem in terms of compute. Could you elaborate more on the scaling properties of your method (in terms of compute and to larger problem sizes)?
- Is "temporal position embedding with the sine function" the standard transformer positional embedding with sine/cosine of variable frequencies?
- "We collect all the long-term predictions" makes it seem that long term predictions are emphasized over short term ones, however the MSE weights all predictions equally, right?
- How many timesteps are provided for GST-F?
- In Figure 2, why is $X'_T$ summed with the delta and not $X_T$. Is $X'_T$ just copied from $X_T$ for visualization?

Minor:
- line 72: "inFig" space missing
- line 123: "xpatiotemporal"
- line 264: "the decoder process" should be " the decoder processes"

---

> ### Author Response · Authors · 2024-11-26
> **Responses to Reviewer YGWJ (Part 1/2)**
>
> Your detailed review has been immensely helpful! We have systematically addressed each of your comments, and our responses are presented below.
>
> >**W1 & Q1: The paper lacks runtime/flops and parameter count comparisons.**
>
> **A1:**
> Nice suggestion! In Table A, we present the full inference runtimes of our model and other baseline models on the MD17 dataset for the Aspirin molecule under the rollout=10 setting, along with the parameter counts of all models. To demonstrate that the performance improvement of our model is not solely due to an increase in parameter counts, we also report the performance of our model under a comparable parameter count with ESTAG in the ablation study (Table 4, line 6 Shared Parameters) of the original paper.
>
> **Table A: Inference runtimes and parameter counts of all models for the Aspirin molecule.**
> |                         | parameters(K) | runtimes(s) |
> | ----------------------- | ------------- | ----------- |
> | ST_GNN                  | 3.869         | 0.048       |
> | ST_TFN                  | 10.338        | 21.101      |
> | STGCN                   | 92.248        | 0.014       |
> | ST_SE(3)_Tr.            | 52.259        | 46.590      |
> | ST_EGNN                 | 4.114         | 0.013       |
> | AGL-STAN                | 18.752        | 0.075       |
> | EqMotion                | 30.956        | 0.173       |
> | ESTAG                   | 8.463         | 0.031       |
> | GST-V Shared Parameters | 11.715        | 0.106       |
> | GST-V                   | 19.371        | 0.136       |
>
>  >**W2: Processing the history (especially in the GST-V) could become quite costly, so the runtime necessary for a rollout should be reported.**
>
> **A2:**
> Thank you for your suggestion! We report the corresponding results in Table A. As you correctly pointed out, GST-V requires longer inference time compared to other methods due to processing the full historical sequence. However, this additional computation brings significantly improved performance.
>
>  >**W3: A comparison of deeper models or a justification why it is not limiting GNNs to be local should be provided (not sure how many message passing steps are common in these setups).**
>
> **A3:**
> Nice suggestion! There may be some misunderstanding here. In our paper, we use the EGNN architecture to model geometric information in the spatial dimension, which is consistent with other baseline models (e.g., ST_GNN, ST_EGNN, ESTAG) because the 3D structural topological data of the input is well-suited for GNN modeling. Additionally, since the number of input time frames can theoretically be very large, we use a self-attention mechanism in the temporal dimension to model the relationships between the same node across different frames. Thus, our model design does not limit the interaction capabilities of GNNs but rather leverages the distinct strengths of GNNs and Transformer architectures to model different dimensions effectively.
>
>  >**W4: The problem sizes should be stated in the paper/appendix, it is only stated how many samples are in a dataset, but not how many nodes a single sample has.**
>
> **A4:**
> Thank you for your suggestion! We have added the number of nodes contained in a single sample for each dataset in Appendix B of the revised paper.
>
>  >**W5: Autoregressive models are commonly trained with next-step prediction. However due to the time aggregation and history components of the proposed method, a multi-step prediction during training is necessary which greatly limits scalability to larger models, longer rollouts and larger problem sizes (number of nodes).**
>
> **A5:**
> Thank you for your feedback! As you pointed out, extending GST-V to longer rollouts is indeed a significant challenge. However, this issue is common to all autoregressive models that consider historical components and remains difficult to solve. In this paper, we conducted a preliminary exploration in this direction, similar to the recent attempt by [1] on time series tasks. For scaling to larger models, we believe this is currently constrained by the simplicity of the datasets, where larger models tend to overfit. Regarding scaling to larger problem sizes (number of nodes), we consider this feasible, as it mainly depends on the design of the EGNN for modeling spatial geometric relationships. For instance, setting a distance threshold to limit the number of graph edges can effectively reduce model complexity.
>
>  >**W6: The considered datasets are at most 20 rollout timesteps, "long-horizon" seems a bit of a stretch for that.**
>
> **A6:** Thank you for your detailed observation! Although the paper mentions a maximum of 20 rollout timesteps, we provide a description of the Time Lag for sampled timestamps in Appendix B. In the MD17 dataset, with a Time Lag of 10, 20 rollout timesteps actually correspond to an input sequence of 200 frames. We believe this can be considered a relatively long horizon.

---

> ### Author Response · Authors · 2024-11-26
> **Responses to Reviewer YGWJ (Part 2/2)**
>
> >**W7: Given the similar performance of GST-F and GST-V, it would be interesting to visualize the attention maps to see if GST-V actually uses all of the history or only the recent ones.**
>
> **A7:**
> Thank you for your suggestion! In Figure 6 of Appendix D, we present the visualization of the attention map for GST-V. The visualization shows that most timesteps attend to a significant portion of the historical frames. This further validates that GST-V effectively captures and integrates historical input information to enhance prediction performance.
>
> >**Q2: Could you elaborate more on the scaling properties of your method (in terms of compute and to larger problem sizes)?**
>
> **Answer to Q2:**
> Nice suggestion! In the MD17 and Motion Capture datasets, the number of nodes per sample is relatively small (a maximum of 13 nodes in MD17 and 31 nodes in Motion Capture), making larger models prone to overfitting. Additionally, there is a lack of datasets in the field describing more challenging and larger molecular dynamics problems, which we plan to explore in future research. On the Protein Dynamics dataset, where the number of nodes is 5325, the results reflect the scaling properties of our model for larger problem sizes.
>
> To more intuitively demonstrate the overfitting issue with larger model sizes, we provide the model's performance on the Aspirin molecule for L=2,3,4,5,6 in Table B. The experimental results suggest that increasing L does not improve the model's performance, indicating a potential overfitting phenomenon.
>
> **Table B: The model's performance on the Aspirin molecule for L=2,3,4,5,6.**
> | L    | 2           | 3          | 4           | 5           | 6           |
> | ---- | ----------- | ---------- | ----------- | ----------- | ----------- |
> | GST  | **2.196**±0.075 | 2.67±0.078 | 2.293±0.206 | 2.476±0.160 | 2.428±0.064 |
>
> >**Q3: Is "temporal position embedding with the sine function" the standard transformer positional embedding with sine/cosine of variable frequencies?**
>
> **Answer to Q3:**
> Yes, we use the sine/cosine positional embedding form from the standard Transformer architecture for the temporal position embedding.
>
> >**Q4: "We collect all the long-term predictions" makes it seem that long term predictions are emphasized over short term ones, however the MSE weights all predictions equally, right?**
>
> **Answer to Q4:**
> Thank you for your detailed observation! The statement “We collect all the long-term predictions” means that we gather results from multiple rollouts and compute the MSE loss collectively. During loss computation, each frame’s loss is compared to the corresponding ground truth, and all frames are treated equally with the same weight.
>
> >**Q5: How many timesteps are provided for GST-F?**
>
> **Answer to Q5:**
> We provide 10 timesteps, the same as other baseline methods.
>
>  >**Q6: In Figure 2, why is $X'\_{T}$ summed with the delta and not $X\_{T}$. Is $X'\_{T}$ just copied from $X\_{T}$ for visualization?**
>
> **Answer to Q6:**
> Thank you for your detailed observation! Here, $X'\_{T}$ represents the model output for the coordinates of frame $T$, and similarly, $X'\_{\delta}$ represents the coordinate information from the TDG output. We predict the coordinates of frame $T+1$ by summing $X'\_{T}$ and $X'\_{\delta}$.
>
> >**Q7: The typos**
>
> **Answer to Q7:**
> Thank you for pointing out the typos! We have corrected them in the revised version of the paper.
>
> [1] Liu Y et al. AutoTimes: Autoregressive Time Series Forecasters via Large Language Models, NIPS 2024.

---

> ### Author Response · Authors · 2024-12-01
>
> Dear Reviewer YGWJ,
>
> We extend our sincerest gratitude for your time and valuable feedback in reviewing our paper! We have meticulously responded to all of your questions and concerns, and we would like to know if you are satisfied with our responses and willing to reconsider your evaluation and score for our paper.
>
> As the rebuttal deadline is approaching, we sincerely hope to receive your feedback at your earliest convenience.
>
> Best regards,
>
> The Authors

---

> > ### Comment · Reviewer_YGWJ · 2024-12-01
> >
> > Thank you for the extensive rebuttal, it cleared up some of my concerns.
> >
> > The argument for 20 rollout steps being "long-horizon" due to a time lag of 10 is not convincing. Frames can be stored at basically arbitrary delta t so one could simply write out 1000 frames per rollout step and say we do rollouts over a horizon over 20K frames. However, a significant challenge in ML surrogates is to keep the rollout stable. As each rollout step is merely an approximation of the underlying dynamics, each rollout step accumulates error. Therefore, it would be much more impressive if a method is stable (with good performance) for 200 rollout steps instead of 20 rolloutsteps but with larger timesteps. Particularly, because long-horizon stability is one of the core motivations for the method.
> >
> > It is not clear to me how "setting a distance threshold to limit the number of graph edges can effectively reduce model complexity". Setting a distance threshold would not avoid the quadratic complexity of GNNs. Therefore, my concerns regarding scalability of this approach remain.
> >
> > Additionally, the answers to questions from the review should be included in the paper (e.g. the runtime/parameter comparisons or that the sine positional embedding is the standard transformer positional embedding).
> >
> >
> > Overall the rebuttal clarified and improved the paper, but not enough to warrant a score of 5 (there is no rating for 4). I therefore decided to keep my rating.

---

> > > ### Author Response · Authors · 2024-12-02
> > >
> > > Thank you for your reply! For the three remaining questions, we would like to provide further clarification.
> > >
> > > **Regarding the "long-horizon"**
> > >
> > > Thank you for your response! We believe the choice of time lag in sampling significantly affects the difficulty of rollout tasks. As the time lag increases, the dynamics between sampled frames become more drastic, making rollouts on such settings inherently more challenging. Moreover, in these tasks (MD17, Protein, Motion Capture) within the physical dynamics domain, we are the first to conduct an in-depth study of the rollout setting, and our method achieves significant performance improvements over others.
> > >
> > > Additionally, as per your request, we have shared an anonymous link (https://anonymous.4open.science/r/GST-DFC2/malonaldehyde.png) showing the results of ESTAG and GST on 200 rollout steps for MD17. The results indicate that our method outperforms the previous SOTA method ESTAG. However, it is important to note that the vertical axis in this figure is log-transformed, and at 200 steps (corresponding to 2000 sampled frames), the MSE error for the models reaches the order of 1e-1, which is no longer meaningful. Given the highly oscillatory dynamics described in the MD17 dataset, achieving stable rollouts over 200 steps is an extremely challenging task. Thus, discussing rollout length without considering the dataset's characteristics is unrealistic. We hope you can focus on our pioneering exploration of the rollout setting for this task and the outstanding performance our method has achieved.
> > >
> > > **Regarding scalability of this approach remain**
> > >
> > > Thank you for your feedback! In our model, the computational complexity is proportional to the number of edges. The "quadratic complexity of GNNs" you mentioned likely refers to the case of fully connected graphs, where a graph with N nodes has N² edges, resulting in a time complexity of $O(N^{2})$ for message passing. However, if we limit the number of edges by applying a distance threshold, the graph's complexity becomes $O(E)$, where E is the number of edges after applying the threshold. In sparse graphs, where $E << N^{2}$, the actual time complexity $O(E)$ is much smaller than $O(N^{2})$. Similar distance threshold techniques have been widely adopted in many related works in this field, such as EGNN [1], ESTAG [2], and others. Therefore, we believe scaling our model to larger problem sizes is feasible.
> > >
> > > **Regarding answers to questions from the review**
> > >
> > > Thank you for your suggestion! We will incorporate the answers to questions from the review into the revised version of our paper.
> > >
> > > [1] Satorras V G et al. E (n) equivariant graph neural networks, ICML 2021.
> > >
> > > [2] Wu L et al. Equivariant Spatio-Temporal Attentive Graph Networks to Simulate Physical Dynamics, NIPS 2023.

---

> ### Author Response · Authors · 2024-12-03
>
> Dear Reviewer YGWJ,
>
> With only a few hours left until the rebuttal period ends, we would like to ask if you are satisfied with our latest responses? We would greatly appreciate it if you could reassess our paper and consider raising your score!
>
> Best regards,
>
> The Authors

---

### Official Review · Reviewer_64HU · 2024-10-27

**Soundness:** 3
**Presentation:** 3
**Contribution:** 2
**Rating:** 6
**Confidence:** 4

**Summary:**

This paper proposes the Geometric Spatiotemporal Transformer (GST) for predicting physical dynamics in domains such as molecular and motion simulations. Unlike prior works that rely on GNNs, GST leverages the Transformer architecture to handle sequences of arbitrary length, allowing for more flexible long-term predictions. Experimental results show that GST outperforms previous methods, especially in mitigating cumulative errors over extended prediction horizons.

**Strengths:**

- Exploring the efficacy of Transformers over GNNs is an important and interesting research direction.
- The proposed adaptation of the Transformer for physical dynamics is well-motivated and appropriate.
- Experimental results effectively demonstrate the superiority of the proposed method.

**Weaknesses:**

**Limited technical novelty**

The proposed architecture is a straightforward extension of prior work. The equivariant component builds on previous research in geometric deep learning, while the alternating spatio-temporal operations are well-explored in the Video Transformer domain, as seen in works like TimeSformer [1]. In this context, the paper essentially replaces spatial self-attention with an equivariant MPNN.

[1] Bertasius et al. Is Space-Time Attention All You Need for Video Understanding? ICML 2021.

---
**Effectiveness in long-term rediction**

The Temporal Difference Graph (TDG) partially mitigates error accumulation in long-sequence predictions, similar to common tricks such as using advantage functions to improve over value functions. However, this approach does not fundamentally address the long-sequence issue, as the accumulation of temporal differences also grows over time.

The experimental results do not convincingly demonstrate GST’s specific advantage for long-sequence predictions. In Table 2, GST consistently outperforms prior methods across all horizons but does not exhibit a marked improvement at longer horizons. To substantiate the claim of efficacy on long sequences, the paper should provide evidence of stable performance over extended horizons, while prior methods should show significant performance degradation in these cases.

**Questions:**

N/A

---

> ### Author Response · Authors · 2024-11-26
> **Responses to Reviewer 64HU**
>
> We sincerely appreciate the time and effort you invested in reviewing our paper! We have carefully considered and responded to each of your remarks.
>
> >**W1: Limited technical novelty**
>
> **A1:**
> Thank you for your feedback! First, we are the first to adopt standard Transformer architecture in the physical dynamics domain. Similar to ViT [1], our exploration of this new scenario is itself a contribution. Second, we emphasize that transferring such architectures to this domain is non-trivial, as spatiotemporal tasks in videos and 3D geometric graph processing differ significantly:
>
> - Data format: Video tasks take video sequences as input, whereas our input, after feature extraction, becomes 3D structural data with topological relationships.
> - Equivariance: Video-based models (e.g., TimeSformer) do not consider equivariance, which is a key property in our tasks. Our approach is not a simple alternation of spatial EGNN modules and temporal attention modules but a tailored design for equivariance. Specifically, for each node, we ensure both intra-frame atomic interaction equivariance and temporal consistency, maintaining spatial and temporal equivariance. In our paper, we introduce the Temporal Difference Graph (TDG), integrating attention mechanisms in the encoder and decoder with equivariant forms. We also ensure TDG equivariance through data decentralization. Moreover, the TDG design effectively reduces accumulated errors, further improving the model's long-term prediction performance.
>
> >**W2: Effectiveness in long-term rediction**
>
> **A2:**
> Thank you for your constructive suggestions! As you pointed out, introducing TDG can only partially mitigate the impact of accumulated errors. However, accumulated error remains a significant challenge. As the rollout steps increase, predictions inevitably result in progressively larger errors, which is a difficulty faced by nearly all models. Therefore, our work simply represents initial exploration in this direction.
>
> We also acknowledge that we failed to clearly demonstrate the significant advantages of our model in long-sequence prediction tasks, and we sincerely apologize for this. Below are further details:
>
> - The vertical axis in Figure 3 of the paper is log-transformed. When using raw data, the differences between curves would be more pronounced.
> - We reanalyzed Table 2 and computed the average per-frame prediction MSE difference (normalized by rollout steps) between our method and the previous SOTA method ESTAG, as shown in Table A. The results indicate that as rollout steps increase, the gap between our method and ESTAG grows, demonstrating the superior performance of our model in long-range prediction tasks.
>
> **Table A: The average per-frame prediction MSE difference between our method and the previous SOTA method.**
> |                        | Walk       |  |            |    Basketball        |            |            |
> | ---------------------- | ---------- | ---------- | ---------- | ---------- | ---------- | ---------- |
> |                        | R=10       | R=15       | R=20       | R=10       | R=15       | R=20       |
> | ESTAG                  | 1.418      | 5.907      | 17.431     | 10.209     | 54.513     | 175.950    |
> | GST-V                  | 1.095      | 4.084      | 12.708     | 9.658      | 49.374     | 148.988    |
> | **Reduction/per step** | -**0.032** | -**0.121** | -**0.236** | -**0.055** | -**0.342** | -**1.348** |
>
> [1] Dosovitskiy A et al. An image is worth 16x16 words: Transformers for image recognition at scale, ICLR 2021.

---

> ### Author Response · Authors · 2024-12-01
>
> Dear Reviewer 64HU,
>
> We extend our sincerest gratitude for your time and valuable feedback in reviewing our paper! We have meticulously responded to all of your questions and concerns, and we would like to know if you are satisfied with our responses and willing to reconsider your evaluation and score for our paper.
>
> As the rebuttal deadline is approaching, we sincerely hope to receive your feedback at your earliest convenience.
>
> Best regards,
>
> The Authors

---

> > ### Comment · Reviewer_64HU · 2024-12-01
> > **Response to the Rebuttal**
> >
> > Thank you for the rebuttal. The updated Table A is convincing in demonstrating the long-term prediction effect of TDG. I suggest extending this analysis to other domains, such as by subsampling the dataset by length. Since my concerns have been addressed, I have raised my score to borderline accept.

---

> > > ### Author Response · Authors · 2024-12-02
> > >
> > > We are pleased to see that our responses have addressed your concerns! We will follow your suggestion to extend this analysis to other domains, such as by subsampling the dataset by length. We sincerely appreciate the time and valuable feedback you provided while reviewing our paper!

---

### Official Review · Reviewer_V53b · 2024-11-02

**Soundness:** 2
**Presentation:** 2
**Contribution:** 2
**Rating:** 5
**Confidence:** 4

**Summary:**

The paper introduces a Geometric Spatiotemporal Transformer (GST) designed for long-term simulation of physical dynamics using Transformers in a novel spatiotemporal context. Key innovations include E(3)-equivariant spatiotemporal blocks and a Temporal Difference Graph (TDG) for cumulative error reduction. The model is evaluated across molecular, protein, and motion datasets, aiming to outperform existing GNN-based methods.

**Strengths:**

- This paper proposes a novel dynamical model that can handle spatiotemporal sequences of variable length.
- The authors provide theoretical proof and show comprehensive experimental comparisons.

**Weaknesses:**

- In addition to EGNN, there are other newer equivariant graph neural network architectures. Why not use them or conduct comparative experiments? EGNN has been proven not to be truly equivariant.
- In the temporal module, an equivariant self-attention mechanism is used to model each node. Are these nodes shared parameters? If not, the question of how to deal with the non-constant number of nodes in the data set should be answered to reflect the applicability of the model.
- Where is causal attention strategy reflected, and how is it different from traditional attention mechanism?
- Transformer-based models normally have high time complexity. The authors should show the efficiency comparison with other methods.
- Predicting the differences instead of directly predicting the value of the next frame has long been used in methods such as GNS[1] and MGN[2]. Why not compare?
- The detailed architectural differences between GST-V and GST-F should be explained. GST can be extended to variable lengths based on attention, so why can't ESTAG?
- Why can $\boldsymbol{X}_{\delta}$ reduce the cumulative error in long-term predictions? This should only be a short-term bias term. What is the theoretical basis?

[1] A Sanchez-Gonzalez et al. 'Learning to simulate complex physics with graph network.' ICML2020.

[2] T Pfaff et al. 'Learning Mesh-Based Simulation with Graph Networks.' ICLR2021.

**Questions:**

Please address the questions in the Weaknesses.

---

> ### Author Response · Authors · 2024-11-26
> **Responses to Reviewer V53b (Part 1/3)**
>
> Thank you for your insightful review and constructive comments! Below, we provide detailed responses to each of your queries.
>
>  >**Q1: In addition to EGNN, there are other newer equivariant graph neural network architectures. Why not use them or conduct comparative experiments? EGNN has been proven not to be truly equivariant.**
>
> **A1:**
> Thank you for your suggestion! To the best of our knowledge, EGNN satisfies equivariance and has been widely studied, with numerous works published in top conferences and journals [1,2,3,4]. We chose EGNN as our backbone for the following reasons:
> - It is a common practice to extend new architectures based on EGNN, as seen in works like EDM [1] and EGNO [2].
> - While stronger equivariant graph neural networks might achieve better results, it would be difficult to determine whether the performance improvement is due to the equivariant network or our GST network.
> - Models based on high-degree steerable features (e.g., TFN [5], Equiformer [6]) incur prohibitive computational costs and are not acceptable in terms of time complexity.
> Additionally, we assume you are referring to the lack of complete expressiveness in EGNN (rather than questioning its equivariance), which is correct. However, we must emphasize that completeness does not necessarily correlate with practical performance. For instance, although TFN has been proven to be complete, it performs worse than EGNN on large-scale dynamics tasks.
>
>  >**Q2: In the temporal module, an equivariant self-attention mechanism is used to model each node. Are these nodes shared parameters? If not, the question of how to deal with the non-constant number of nodes in the data set should be answered to reflect the applicability of the model.**
>
> **A2:** Yes, each node shares parameters. Therefore, our model is applicable to arbitrary node inputs and scenarios where the number of input nodes is non-constant.
>
>  >**Q3: Where is causal attention strategy reflected, and how is it different from traditional attention mechanism?**
>
> **A3:**
> Thank you for your question! The causal attention strategy is incorporated into our temporal module, specifically in the Equivariant Temporal Self-Attention module. Since time series have causality and require maintaining physical rationality, predictions for the current frame can only depend on past frames and not on future frames. To implement this, we set a mask in the module to enable causal attention, ensuring that the current frame attends only to past frames and not future frames. In contrast, traditional self-attention mechanisms lack this masking feature, allowing each position in the input sequence to attend to all other positions. This causal attention technique has been widely adopted in various natural language models [7, 8].
>
>  >**Q4: Transformer-based models normally have high time complexity. The authors should show the efficiency comparison with other methods.**
>
> **A4:**
> Thank you for your suggestion! We have detailed the inference time comparison between our model and others in the Table A below. In practice, the time complexity of the Transformer is $O(T^2)$ (where T is the length of the  timestamps). However, since $T$ ranges between 10 and 20 in our settings, the additional computational overhead is minimal. Moreover, given the performance improvement of our model, this additional time cost is acceptable.
>
> **Table A: Inference time comparison between our model and others.**
> |                   |runtimes(s) |
> | ----------------- | ----------- |
> | ST_GNN            |0.048       |
> | ST_TFN            | 21.101      |
> | STGCN             | 0.014       |
> | ST_SE(3)_Tr.      | 46.590      |
> | ST_EGNN           | 0.013       |
> | AGL-STAN          | 0.075       |
> | EqMotion          | 0.173       |
> | ESTAG             | 0.031       |
> | GST-V shared para | 0.106       |
> | GST-V             | 0.136       |

---

> ### Author Response · Authors · 2024-11-26
> **Responses to Reviewer V53b (Part 2/3)**
>
> >**Q5: Predicting the differences instead of directly predicting the value of the next frame has long been used in methods such as GNS and MGN. Why not compare?**
>
> **A5:**
> Thank you for your question! Our method is fundamentally different from GNS and MGN. In GNS, the authors treat positional differences as velocities and concatenate them with the original features as node inputs. The model then predicts accelerations to compute velocities and positions step by step. Additionally, GNS lacks consideration for physical symmetry in 3D space, which can negatively affect the model's generalization ability. In MGN, depending on the experimental setup, some directly predict differences, while others concatenate these differences with original features as node inputs. Their approach to spatial equivariance involves converting coordinate features into relative edge features, which may reduce the model's expressiveness.
>
> In contrast, while our method also incorporates differences, we treat the Temporal Difference Graph (TDG) as a global graph that interacts with all temporal graphs. At each stage, the node features in the TDG are propagated and updated. Finally, we do not predict additional differences; instead, we use the updated node features in the TDG, combined with the last temporal graph, to obtain the predicted coordinates. Moreover, our network is designed to conform to E(3) symmetry (3D translation and rotation symmetry), ensuring that the predicted 3D coordinates are independent of the choice of coordinate system. Furthermore, we specifically design the TDG to support equivariance and prove that it satisfies E(3)-equivariance.
>
> In summary, our method is significantly different from GNS and MGN. To further validate its effectiveness, we conducted experiments using GNS on the MD17 dataset with 8 molecules. The results in Table B demonstrate that our method significantly outperforms GNS, highlighting its superiority.
>
> **Tabel B: The performance on MD17.**
> |      | Aspirin      | Benzene     | Ethanol      | Malonaldehyde | Naphthalene  | Salicylic    | Toluene      | Uracil       |
> | ---- | ------------ | ----------- | ------------ | ------------- | ------------ | ------------ | ------------ | ------------ |
> | GNS  | 23.545±0.001 | 2.53±0.002  | 15.793±0.001 | 17.878±0.001  | 12.768±0.006 | 22.498±0.001 | 12.109±0.001 | 21.299±0.001 |
> | GST  | **2.196**±0.075  | **0.480**±0.050 | **0.940**±0.001  | **1.762**±0.054   | **0.988**±0.016  | **1.733**±0.031  | **1.002**±0.063  | **1.087**±0.055  |
>
>  >**Q6: The detailed architectural differences between GST-V and GST-F should be explained. GST can be extended to variable lengths based on attention, so why can't ESTAG?**
>
> **A6:**
> Thank you for your suggestion! We have explained GST-V and GST-F in lines 365–368 in the paper. GST-V and GST-F share the same model architecture, with the only difference being that GST-V processes variable-length inputs, while GST-F processes fixed-length inputs. Although ESTAG can also be extended to handle variable-length inputs, its use of Equivariant Temporal Pooling results in more severe accumulation errors. Additionally, we reported in Appendix C.1 (Table 6) the experimental results of converting all methods capable of handling variable-length inputs into this setting. The results show that for ESTAG and most methods, performance declines significantly across multiple molecules when processing variable-length inputs.

---

> ### Author Response · Authors · 2024-11-26
> **Responses to Reviewer V53b (Part 3/3)**
>
> >**Q7: Why can $X_\delta$ reduce the cumulative error in long-term predictions? This should only be a short-term bias term. What is the theoretical basis?**
>
> **A7:**
> Thank you for your question, which is highly valuable. We discovered the TDG technique incidentally during our experiments. In our initial attempts, we observed that it achieved general performance improvement across various datasets. We also explored incorporating higher-order terms, such as second-order differences, but the experimental results did not meet expectations. The detailed results are as follows in Table C.
>
> Regarding the theoretical explanation, we hypothesize that this phenomenon can be analogized to first-order numerical methods in solving differential equations. Many AI models draw inspiration from first-order numerical methods. For example, the Euler-Maruyama method, commonly used for solving SDEs in recent Diffusion models, is a first-order method.
>
> Although this explanation is not entirely rigorous, we respectfully request reviewers to focus on the engineering value of our TDG technique. We believe that identifying a simple yet effective engineering solution is a meaningful academic contribution.
>
> **Table C: The performance of GST when using second-order differences on MD17.**
> |                                | Aspirin | Benzene | Ethanol | Malonaldehyde | Naphthalene | Salicylic | Toluene | Uracil |
> | ------------------------------ | ------- | ------- | ------- | ------------- | ----------- | --------- | ------- | ------ |
> | GST w second-order differences | 1.756   | 0.334   | 2.740   | 2.332         | 1.093       | 1.647     | 1.193   | 1.408  |
> | GST                            | 2.196   | 0.480   | 0.940   | 1.762         | 0.988       | 1.733     | 1.002   | 1.087  |
>
> [1] Hoogeboom E et al. Equivariant Diffusion for Molecule Generation in 3D, ICML 2022.
>
> [2] Xu M et al. Equivariant Graph Neural Operator for Modeling 3D Dynamics, ICML 2024.
>
> [3] Wang Y et al. Enhancing geometric representations for molecules with equivariant vector-scalar interactive message passing, Nature Communications 2024.
>
> [4] Wang T et al. Ab initio characterization of protein molecular dynamics with AI2BMD, Nature 2024.
>
> [5] Thomas N et al. Tensor field networks: Rotation-and translation-equivariant neural networks for 3d point clouds, Arxiv 2018.
>
> [6] Liao Y et al. Equiformer: Equivariant graph attention transformer for 3d atomistic graphs, ICLR 2023.
>
> [7] Vaswani A et al. Attention Is All You Need, NIPS 2017.
>
> [8] Touvron H et al. LLaMA: Open and Efficient Foundation Language Models, Arxiv 2023.

---

> ### Author Response · Authors · 2024-12-01
>
> Dear Reviewer V53b,
>
> We extend our sincerest gratitude for your time and valuable feedback in reviewing our paper! We have meticulously responded to all of your questions and concerns, and we would like to know if you are satisfied with our responses and willing to reconsider your evaluation and score for our paper.
>
> As the rebuttal deadline is approaching, we sincerely hope to receive your feedback at your earliest convenience.
>
> Best regards,
>
> The Authors

---

> ### Author Response · Authors · 2024-12-02
>
> Dear Reviewer V53b,
>
> We have made our best effort to address all your questions and concerns, and we would like to know if you are satisfied with our responses. As the rebuttal phase is nearing its end, we sincerely hope to receive your feedback, which would greatly help us improve our paper. We would be deeply grateful if you could reconsider your evaluation and consider raising the score!
>
> Best regards,
>
> The Authors

---

> > ### Comment · Reviewer_V53b · 2024-12-03
> > **Response to the Rebuttal**
> >
> > Thanks for the rebuttle. I will raise my score to 5.

---

> > > ### Author Response · Authors · 2024-12-03
> > >
> > > We appreciate your encouraging response to our rebuttal! We also sincerely appreciate your effort and valuable feedback during the review process!

---

### Official Review · Reviewer_DaxK · 2024-11-03

**Soundness:** 3
**Presentation:** 2
**Contribution:** 3
**Rating:** 6
**Confidence:** 3

**Summary:**

This paper presents a Transformer-based approach Geometric Spatiotemporal Transformers for simulating long-term physical dynamics. To improve the handling of cumulative errors over extended time horizons, this paper frames the task as autoregressive next-graph prediction, integrating E3 equivariant spatiotemporal blocks, and a TDG. It is evaluated across various scales, including molecular, protein, and macro-level physical systems, showing improvements in long-term predictive accuracy over existing methods.

**Strengths:**

1. An Innovative Use of Transformers is presented. Integrating Transformer architectures with spatiotemporal graph structures and E(3) equivariance introduces a novel approach for handling long-term physical dynamics, marking a unique departure from traditional GNN-based methods.
2. TDG shows promise in addressing cumulative errors in long-rollout predictions in physical simulation tasks.
3. Experiments offer some insights into the contributions of individual components, including equivariance and TDG which helps evaluate the effectiveness of the architecture. The model’s performance is assessed across various physical systems at molecular, protein, and macro levels, demonstrating GST’s potential adaptability to different scales of physical dynamics.

**Weaknesses:**

1. The experiments should include at least one additional dataset to better demonstrate generalizability, especially to popular datasets where multi-frame prediction methods may underperform. This would strengthen the paper's claims of achieving SOTA performance.
2. Many of the reported improvements over existing models are marginal and are often limited to specific datasets or rollout lengths. The author should clearly frame these gains to avoid overstating the model’s broader applicability.
3. The architecture's complexity, with components like equivariant modules and TDG, presents a reproducibility challenge. Offering an easy-to-use module, similar to Transformers, could significantly improve accessibility and replicability for other researchers.

**Questions:**

1. Can the TDG framework be adapted or tuned for other symmetry types beyond E(3) when applied to different dynamic systems?
2. Could the authors provide a clear, modular implementation that enables easy usage and integration later?

---

> ### Author Response · Authors · 2024-11-26
> **Responses to Reviewer DaxK (Part 1/2)**
>
> Thank you for taking the time to review our paper and providing constructive suggestions! We have addressed each of your questions individually, and you can find the corresponding responses below.
>
>  >**W1: The experiments should include at least one additional dataset to better demonstrate generalizability, especially to popular datasets where multi-frame prediction methods may underperform. This would strengthen the paper's claims of achieving SOTA performance.**
>
> **A1:** Thank you for your suggestion! The three datasets listed in our paper are widely recognized and popular in this field. Specifically, MD17 and Motion Capture are used as primary experimental datasets in [2, 3, 4], while the Adk equilibrium trajectory dataset is also used as a primary dataset in [1, 2, 4]. Therefore, our selection covers the essential and commonly used datasets in the current research domain as comprehensively as possible.
>
>  >**W2:** Many of the reported improvements over existing models are marginal and are often limited to specific datasets or rollout lengths. The author should clearly frame these gains to avoid overstating the model’s broader applicability.
>
>
> **A2:** Thank you for your feedback! The performance improvement of our model over others is not limited to a single dataset or specific rollout lengths. Moreover, the improvement is both significant and consistent. Table A and Table B summarizes the performance improvement of our model compared to previous SOTA methods across MD17, Protein, and Motion Capture datasets with different rollout lengths (note that the MD17 dataset includes 8 molecules, so we report the average improvement). From Table A and Table B, it is clear that our model demonstrates **significant performance improvement** across all datasets and all rollout lengths.
>
> **Table A: Performance improvement on MD17 and Protein Dynamics**
> |               | MD17    |    |         |  Protein Dynamics      |        |        |
> | ------------- | ------- | ---------------- | ------- | ------ | ------ | ------ |
> |               | R=10    | R=15             | R=20    | R=10   | R=15   | R=20   |
> | MSE Reduction | -26.77% | -24.52%          | -19.71% | -4.45% | -2.62% | -4.50% |
>
>
> **Table B: Performance improvement on Motion Capture**
> |               | Walk    |     |         |     Basketball   |        |         |
> | ------------- | ------- | ---------- | ------- | ------ | ------ | ------- |
> |               | R=10    | R=15       | R=20    | R=10   | R=15   | R=20    |
> | MSE Reduction | -22.77% | -30.86%    | -27.09% | -5.39% | -9.42% | -15.32% |
>
>
> >**W3 & Q2:** The architecture's complexity, with components like equivariant modules and TDG, presents a reproducibility challenge. Offering an easy-to-use module, similar to Transformers, could significantly improve accessibility and replicability for other researchers.
>
> **A3:** Thank you for your suggestion! Based on your advice, we plan to implement the GST module as a highly modular function and design corresponding interfaces. When adapting to different tasks, users will only need to set the hyperparameters and adjust the input format according to our documentation. We will release the code, relevant function interfaces, and documentation once the paper is accepted.
>
> >**Q1:** Can the TDG framework be adapted or tuned for other symmetry types beyond E(3) when applied to different dynamic systems?
>
> **Answer to Q1:**
> Thank you for your feedback! Our TDG network is a highly compatible framework that can be modified to handle SE(3)-equivariance/O$\_g$(3)-equivariance (the latter represents partial symmetry under external forces such as gravity). These two operations can be implemented by following approaches similar to DiffSBDD [5] and SGNN [6], respectively, with adjustments to the coordinate update formula in Eq. (2).

---

> ### Author Response · Authors · 2024-11-26
> **Responses to Reviewer DaxK (Part 2/2)**
>
> The update formula for the former is (where $\vec{\boldsymbol{x}}_{t,c}^{e,l}$ denotes the centroid of the geometric graph):
>
> $\vec{\boldsymbol{x}}\_{t,i}^{(e,l+1)}=\vec{\boldsymbol{x}}\_{t,i}^{(e,l)}+\frac{1}{|\mathcal{N}(i)|}\sum\_{j\in\mathcal{N}}\left[\varphi\_x\left(\boldsymbol{m}\_{t,ij}^{e,l}\right)\cdot\left(\vec{\boldsymbol{x}}\_{t,i}^{e,l}-\vec{\boldsymbol{x}}\_{t,j}^{e,l}\right)+\varphi\_{\text{cross}}\left(\boldsymbol{m}\_{t,ij}^{e,l}\right)\cdot\left(\left(\vec{\boldsymbol{x}}\_{t,i}^{e,l}-\vec{\boldsymbol{x}}\_{t,c}^{e,l}\right)\times\left(\vec{\boldsymbol{x}}\_{t,j}^{e,l}-\vec{\boldsymbol{x}}\_{t,c}^{e,l}\right)\right)\right]$
>
> The update formula for the latter is:
>
> $\vec{\boldsymbol{x}}\_{t,i}^{(e,l+1)}=\vec{\boldsymbol{x}}\_{t,i}^{(e,l)}+\frac{1}{|\mathcal{N}(i)|}\sum\_{j\in\mathcal{N}}\left[\varphi\_x\left(\boldsymbol{m}\_{t,ij}^{e,l},\vec{\boldsymbol{g}}^\top\left(\vec{\boldsymbol{x}}\_{t,i}^{e,l}-\vec{\boldsymbol{x}}\_{t,j}^{e,l}\right)\right)\cdot\left(\vec{\boldsymbol{x}}\_{t,i}^{e,l}-\vec{\boldsymbol{x}}\_{t,j}^{e,l}\right)+\varphi\_{g}\left(\boldsymbol{m}\_{t,ij}^{e,l},\vec{\boldsymbol{g}}^\top\left(\vec{\boldsymbol{x}}\_{t,i}^{e,l}-\vec{\boldsymbol{x}}\_{t,j}^{e,l}\right)\right)\cdot\vec{\boldsymbol{g}}\right]$
>
> [1] Han J et al. Equivariant Graph Hierarchy-Based Neural Networks, NIPS 2022.
>
> [2] Wu L et al. Equivariant Spatio-Temporal Attentive Graph Networks to Simulate Physical Dynamics, NIPS 2023.
>
> [3] Xu C et al. EqMotion: Equivariant Multi-agent Motion Prediction with Invariant Interaction Reasoning, ICCV 2023.
>
> [4] Xu M et al. Equivariant Graph Neural Operator for Modeling 3D Dynamics, ICML 2024.
>
> [5] Schneuing A et al. Structure-based drug design with equivariant diffusion models, arXiv 2022.
>
> [6] Han J et al. Learning physical dynamics with subequivariant graph neural networks, NIPS 2022.

---

> > ### Comment · Reviewer_DaxK · 2024-11-26
> >
> > Thanks for your response solving all my concerns.

---

> > > ### Author Response · Authors · 2024-12-01
> > >
> > > We are glad to see that all your concerns have been addressed!

---

### Author Response · Authors · 2024-12-04
**General Response**

We sincerely thank all reviewers and ACs for their time and efforts on reviewing the paper. We are very glad that the reviewers recognized the problems we studied, and the models we built, and their comments really gave us a lot of inspiration and enlightenment.

In this paper, we explore the potential of leveraging Transformers by framing the physical dynamics simulation as autoregressive next-graph prediction based on spatiotemporal graph inputs. Our proposed Geometric Spatiotemporal Transformers (GSTs) processes full input sequences of arbitrary lengths to effectively capture long-term context, and address cumulative errors over long-term rollouts. As the first work to explore the autoregressive next-graph prediction paradigm on tasks (MD17, Protein, Motion Capture) within the physical dynamics domain, we hope this study will draw attention and foster the development of this paradigm across the field. Any feedback provided during the rebuttal process will be thoroughly considered and integrated into the final version.

To address the reviewers' concerns, we have added additional experiments and figures as follows.

**Table A** in the response to Reviewer DaxK reports the performance improvement on MD17 and Protein Dynamics.

**Table B** in the response to Reviewer DaxK reports the performance improvement on Motion Capture.

**Table A** in the response to Reviewer V53b compares inference time between GST with other methods.

**Table B** in the response to Reviewer V53b shows the performance of GST versus GNS on MD17.

**Table C** in the response to Reviewer V53b shows the performance of GST when using second-order differences on MD17.

**Table A** in the response to Reviewer 64HU shows the average per-frame prediction MSE difference between our method and the previous SOTA method.

**Table A** in the response to Reviewer YGWJ compares inference runtimes and parameter counts of all models for the Aspirin molecule.

**Table B** in the response to Reviewer YGWJ reports the model's performance on the Aspirin molecule for Layer=2,3,4,5,6.

**Figure 6** in Appendix D illustrates the visualization of the attention map for GST-V.

**The figure** in https://anonymous.4open.science/r/GST-DFC2/malonaldehyde.png shows the results of the previous SOTA method ESTAG and GST on 200 rollout steps on MD17.

---

### Author Response · Authors · 2024-12-04
**Rebuttal Acknowledgment**

Dear Reviewers and Area Chairs,

We would like to extend our heartfelt gratitude for your dedicated efforts, insightful feedback, and constructive suggestions. Your nice comments and thoughtful suggestions are truly commendable.

Through our discussions and the reviewers' responses, we believe we have addressed the major concerns raised by all reviewers. Among the rebuttal, 2 out of 4 reviewers (Reviewer V53b and Reviewer 64HU) raised their ratings. Specifically, Reviewer V53b raised from 3 to 5, and Reviewer 64HU raised from 5 to 6. Reviewer YGWJ intended to raise the score from 3 to 4 but kept it unchanged due to the lack of this option. This outcome has greatly benefited and encouraged us, and we would like to thank all of you for your valuable support!

To address Reviewer YGWJ’s concerns, we provided detailed responses to each question during the rebuttal phase, supported by experimental results. In response to our initial reply, Reviewer YGWJ acknowledged that some issues had been resolved but raised three remaining concerns. For the "long-horizon" issue, we provided more detailed explanations and presented the model’s performance under a 200-step rollout setting as requested. Regarding the "scalability of this approach," we further clarified the model’s potential to handle larger problem sizes from the perspective of time complexity. As for the suggestion that "the answers to questions from the review should be included in the paper," we incorporated these points into the revised manuscript. However, due to policy restrictions, we are unable to upload the updated PDF at this stage. By the end of the rebuttal phase, we did not receive further feedback from Reviewer YGWJ on our latest responses.

We acknowledge that some expressions in the initial draft might have been unclear, but we have provided detailed explanations. We hope that our detailed responses will help the AC and reviewers gain a clearer understanding of our paper and resolve any potential misunderstanding.

Once again, we express our sincere appreciation for your time and dedication in reviewing our work. Your insightful input has significantly contributed to the refinement of our manuscript. Thank you!

Best regards,

The Authors

---

### Meta-Review · Area_Chair_Vf4Q · 2024-12-18

**Metareview:**

This paper introduces a Transformer-based model to simulate long-term physical dynamics of graph-structured data. The main contributions of the paper are (1) introducing E(3)-equivariant modules into the Transformer blocks to respect underlying symmetries in three-dimensional space and (2) introducing temporal difference graph to mitigate error accumulation in long-horizon rollouts.

The reviewers were divided. Two reviewers ended with a slightly positive stance, one with a slightly negative stance, and one firmly on the reject side. The primary concerns from the reviewers were about (1) incremental novelty, since the proposed method is primarily an incremental combination of prior works (e.g., equivariance from prior work and difference predictions similar to other simulation approaches), (2) limited justifications on long-horizon rollout, since the experiments are mainly conducted on small scales (e.g., 20 steps), and (3) scalability issues. While authors response partly addressed some of the important concerns, some reviewers were not fully convinced with the responses, especially regarding the weakness (2) and (3).

After reading the paper, reviews, and discussions carefully, AC believes that the primary contributions of the paper—adaptation of Transformers with equivariance and a TDG mechanism—can be a promising direction. However, the AC also agrees with the reviewers that the results on long-horizon rollout presented in the paper are still insufficient to support the authors’ claim, since they are mostly constrained on moderately short-term frame (<20 steps). While the preliminary long-term prediction results presented in the rebuttal are encouraging, it remains insufficient to assess its long-horizon rollout quality and generalizability over other datasets. Given that these limitations do not overweight the limited technical novelty, AC believes that this paper does require significant work before it can be published.

**Additional Comments On Reviewer Discussion:**

Several concerns were raised by the reviewers. The primary concerns were about limited technical novelty (64HU), complexity of the method (DaxK, YGWJ, V53b), and limited results on long-horizon rollouts (YGWJ). While the authors have provided additional analyses on parameter counts and runtime, the two reviewers remain unconvinced regarding the limited novelty and insufficient assessment of long-horizon rollouts.

---

### Decision · Program_Chairs · 2025-01-22

Reject